# Edit-Based Flow Matching for Temporal Point Processes

**David Lüdke**[*]**, Marten Lienen**[*]**, Marcel Kollovieh, Stephan Günnemann**
{d.luedke,m.lienen,m.kollovieh,s.guennemann}@tum.de
School of Computation, Information and Technology & Munich Data Science Institute
Technical University of Munich

## Abstract

Temporal point processes (TPPs) are a fundamental tool for modeling event sequences in continuous time, but most existing approaches rely on autoregressive parameterizations that are limited by their sequential sampling. Recent non-autoregressive, diffusion-style models mitigate these issues by jointly interpolating between noise and data through event insertions and deletions in a discrete Markov chain. In this work, we generalize this perspective and introduce an Edit Flow process for TPPs that transports noise to data via insert, delete, and substitute edit operations. By learning the instantaneous edit rates within a continuous-time Markov chain framework, we attain a flexible and efficient model that effectively reduces the total number of necessary edit operations during generation. Empirical results demonstrate the generative flexibility of our unconditionally trained model in a wide range of unconditional and conditional generation tasks on benchmark TPPs.

## 1 Introduction

Temporal point processes (TPPs) capture the distribution over sequences of events in time, where both the continuous arrival-times and number of events are random. They are widely used in domains such as finance, healthcare, social networks, and transportation, where understanding and forecasting event dynamics and their complex interactions is crucial. Most (neural) TPPs capture the complex interactions between events *autoregressively*, parameterizing a conditional intensity/density of each event given its history (Daley & Vere-Jones, 2006; Shchur et al., 2021). While natural and flexible, this factorization comes with inherent limitations: sampling scales linearly with sequence length, errors can compound in multi-step generation, and conditional generation is restricted to forecasting tasks.

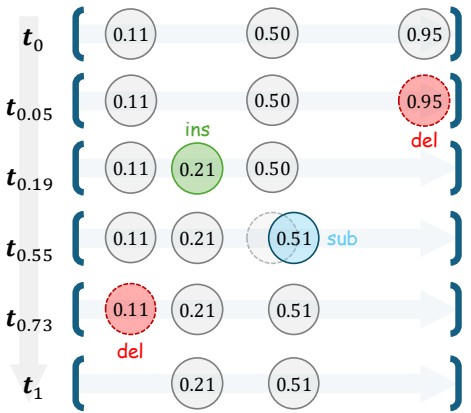

Figure 1: Edit process transporting $t_0 \sim p_{\text{noise}}(t)$ to $t_1 \sim q_{\text{target}}(t)$ by inserting, deleting and substituting events.

**Beyond autoregression.** Recent advances demonstrate that modeling event sequences *jointly* proposes a sound alternative to overcome these limitations. Inspired by diffusion, AddThin (Lüdke et al., 2023) and PSDiff (Lüdke et al., 2025) leverage the thinning and superposition properties of TPPs to construct a discrete Markov chain that learns to transform noise sequences $t_0 \sim p_{\text{noise}}(t)$ into data sequences $t_1 \sim q_{\text{target}}(t)$ through *insertions* and *deletions* of events. These methods highlight the promise of joint sequence modeling for TPPs by learning stochastic set interpolations and have shown state-of-the-art results, especially in forecasting.

In parallel, Havasi et al. (2025) introduced Edit Flow, a discrete flow-matching framework (Gat et al., 2024; Campbell et al., 2024; Shi et al., 2025) for variable-length sequences of tokens (e.g.,

---

[*]Equal contribution
Find the code at cs.cit.tum.de/daml/editpp

language). Their approach models discrete flows in sequence space through *insertions*, *deletions*, and *substitutions*, formalized as a continuous-time Markov Chain (CTMC). To make the learning process tractable, they introduce an expanded auxilliary state space that aligns sequences, simultaneously reducing the complexity of marginalizing over possible transitions and enabling efficient element-wise parameterization in sequence space.

In this paper, we unify these perspectives and propose EDITPP, an Edit Flow for TPPs that learns to transport noise sequences $\boldsymbol{t}_0 \sim p_{\text{noise}}(\boldsymbol{t})$ to data sequences $\boldsymbol{t}_1 \sim q_{\text{target}}(\boldsymbol{t})$ via atomic *edit operations* insertions, deletions, and substitutions (see figure 1). We define these operations specifically for TPPs, efficiently parameterize their instantaneous rates within a CTMC, propose an auxiliary alignment space for TPPs, and show that our unconditionally trained model can be flexibly applied to both unconditional and conditional tasks with adaptive complexity. Our main contributions are:

- We introduce EDITPP, the first generative framework that models TPPs via continuous-time edit operations, unifying stochastic set interpolation methods for TPPs with Edit Flows for discrete sequences.

- We propose a tractable parameterization of insertion, deletion, and substitution rates for TPPs within the CTMC framework, effectively reducing the number of edit operations for generation.

- We demonstrate empirically that EDITPP achieves state-of-the-art results in both unconditional and conditional tasks across diverse real-world and synthetic datasets.

## 2 BACKGROUND

### 2.1 TEMPORAL POINT PROCESSES

TPPs (Daley & Vere-Jones, 2006; 2007) are stochastic processes whose realizations are almost surely finite, ordered sets of random events in time. Let $\boldsymbol{t} = \{t^{(i)}\}_{i=1}^n$, with $t^{(i)} \in [0, T]$, denote a realization of $n$ events on a bounded time interval, which can equivalently be represented by the *counting process* $N(t) = \sum_{i=1}^n \mathbf{1}\{t^{(i)} \leq t\}$ counting the number of events up to time $t$. A TPP is uniquely characterized by its *conditional intensity function* (Rasmussen, 2018):

$$\lambda^*(t) = \lim_{\Delta t \downarrow 0} \frac{\mathbb{E}[N(t + \Delta t) - N(t) \mid \mathcal{H}_t]}{\Delta t}, \tag{1}$$

where $\mathcal{H}_t = \{t^{(i)} : t^{(i)} < t\}$ denotes the history up to time $t$. Intuitively, $\lambda^*(t)$ represents the instantaneous rate of events given the past. Two important properties of TPPs are superposition and thinning. Superposition, i.e., *inserting* one sequence into another, $\boldsymbol{t} = \boldsymbol{t}_1 \cup \boldsymbol{t}_2$, where $\boldsymbol{t}_1$ and $\boldsymbol{t}_2$ are realizations from TPPs with intensities $\lambda_1$ and $\lambda_2$, results in a sample from a TPP with intensity $\lambda = \lambda_1 + \lambda_2$. Independent thinning, i.e., randomly *deleting* any event of a sequence from a TPP with intensity $\lambda$ with probability $p$, results in an event sequence from a TPP with intensity $(1 - p)\lambda$.

The likelihood of observing an event sequence $\boldsymbol{t}$ given the conditional intensity/density is:

$$p(\boldsymbol{t}) = \left(\prod_{i=1}^n p(t^{(i)} \mid \mathcal{H}_{t^{(i)}})\right) (1 - F(T \mid \mathcal{H}_{t^{(i)}})) = \left(\prod_{i=1}^n \lambda^*(t^{(i)})\right) \exp\left(-\int_0^T \lambda^*(s) \mathrm{d}s\right), \tag{2}$$

where $F(T \mid \mathcal{H}_t)$ is the CDF of the conditional event density $p(t \mid \mathcal{H}_t)$. While this autoregressive formulation of TPPs provides a natural framework for modeling event dependencies, it also poses challenges. Parameterizing the conditional intensity or density is generally nontrivial, and the inherently sequential factorization can lead to inefficient sampling, error accumulation, and limits conditional tasks to forecasting (Lüdke et al., 2023; 2025).

### 2.2 MODELING TPPs BY SET INTERPOLATION

Instead of explicitly modeling the intensity function, Lüdke et al. (2023; 2025) leverage the thinning and superposition properties of TPPs to derive diffusion-like generative models that interpolate between data event sequences $\boldsymbol{t}_1 \sim q_{\text{target}}(\boldsymbol{t})$ and noise $\boldsymbol{t}_0 \sim p_{\text{noise}}(\boldsymbol{t})$ by inserting and deleting

elements. ADDTHIN (Lüdke et al., 2023) defines the noising Markov chain recursively over a fixed number of steps with size $\Delta$ indexed by $s \in [0, 1]$ as follows:

$$\lambda_s(t) = \underbrace{\alpha_s \lambda_{s-\Delta}(t)}_{\text{(i) Thin}} + \underbrace{(1 - \alpha_s)\lambda_0(t)}_{\text{(ii) Add}}, \tag{3}$$

where $\lambda_1(t)$ is the unknown target intensity of the TPP and $\alpha_s \in (0, 1)$. Intuitively, this noising process increasingly deletes events from the data sequence, while inserting events from a noise TPP $\lambda_0(t)$. PSDIFF (Lüdke et al., 2025) further separates the adding and thinning to yield a Markov chain for the forward process, that stochastically interpolates between $\boldsymbol{t}_0$ and $\boldsymbol{t}_1$ as follows:

$$p_s(\boldsymbol{t} \mid \boldsymbol{t}_1, \boldsymbol{t}_0) = \prod_{t \in \boldsymbol{t}} \begin{cases} \bar{\alpha}_s & \text{if } t \in \boldsymbol{t}_1 \\ 1 - \bar{\alpha}_s & \text{if } t \in \boldsymbol{t}_0 \end{cases} \tag{4}$$

or equivalently $\lambda_s(t) = \bar{\alpha}_s \lambda_1(t) + (1 - \bar{\alpha}_s)\lambda_0(t)$, with $\bar{\alpha}_s$ being the product of $\alpha_i$'s. Eq. (4) defines an element-wise conditional path by independent insert and delete operations on TPPs, assuming $\boldsymbol{t}_0 \cap \boldsymbol{t}_1 = \emptyset$.

## 2.3 FLOW MATCHING WITH EDIT OPERATIONS

Havasi et al. (2025) introduce Edit Flows, a non-autoregressive generative framework for variable-length token sequences with a fixed, discrete vocabulary (e.g., language). They propose a discrete flow that transports a noisy sequence $\boldsymbol{x}_0 \sim p_{\text{noise}}(\boldsymbol{x})$ to a data sequence $\boldsymbol{x}_1 \sim q_{\text{data}}(\boldsymbol{x})$ via elementary *edit operations*: insertions, deletions, and substitutions. This is formalized via the discrete flow matching framework (Gat et al., 2024; Campbell et al., 2024; Shi et al., 2025) in an augmented space, yielding a CTMC $\Pr(X_{s+h} = \boldsymbol{x} \mid X_s = \boldsymbol{x}_s) = \delta_{\boldsymbol{x}_s}(\boldsymbol{x}) + h u_s^\theta(\boldsymbol{x} \mid \boldsymbol{x}_s) + o(h)$ with transition rates $u_s^\theta$ governed by the edit operations.

Directly defining a conditional rate $u_s(\boldsymbol{x}|\boldsymbol{x}_1, \boldsymbol{x}_0)$ to match $u_s^\theta$ to, as in discrete flow matching, is very hard or even intractable, since all possible edits producing $\boldsymbol{x}$ must be considered. Thus, to train this CTMC, they rely on two major insights. First, a CTMC in a data space $\mathcal{X}$ can be learned by introducing an augmented space $\mathcal{X} \times \mathcal{Z}$ where the true dynamics are known. Second, designing the auxiliary space $\mathcal{Z}$ to follow the element wise mixture probability path $p_s(\boldsymbol{z} \mid \boldsymbol{z}_0, \boldsymbol{z}_1) = \prod_n \left[ (1 - \kappa_s)\delta_{z_0^{(i)}}(z^{(i)}) + \kappa_s \delta_{z_1^{(i)}}(z^{(i)}) \right]$ with kappa schedule $\kappa_s \in [0, 1]$ (Gat et al., 2024) enables training the CTMC directly in the data space $\mathcal{X}$ of variable-length sequences.

Edit operations are encoded by introducing a blank token $\epsilon$ and mapping $(\boldsymbol{x}_0, \boldsymbol{x}_1)$ into aligned sequences $(\boldsymbol{z}_0, \boldsymbol{z}_1)$ in $\mathcal{Z}$, where pairs $(z_0^{(i)}, z_1^{(i)})$ correspond to insertions $(\epsilon, x)$, deletions $(x, \epsilon)$, or substitutions $(x, y)$. Crucially, since the discrete flow matching dynamics in $\mathcal{Z}$ are known, they can be transferred back to $\mathcal{X}$ via $p_s(\boldsymbol{x}, \boldsymbol{z} \mid \boldsymbol{z}_0, \boldsymbol{z}_1) = p_s(\boldsymbol{z} \mid \boldsymbol{z}_0, \boldsymbol{z}_1)\delta_{\text{f}_{\text{rm-blanks}}(\boldsymbol{z})}(\boldsymbol{x})$, by removing $\epsilon$'s with $\text{f}_{\text{rm-blanks}}$. Then, the marginal rates $u_s^\theta$ are learned in $\mathcal{X}$ by marginalizing over $\boldsymbol{z}$ with the Bregman divergence

$$\mathcal{L} = \mathbb{E}_{\substack{(\boldsymbol{z}_0, \boldsymbol{z}_1) \sim \pi(\boldsymbol{z}_0, \boldsymbol{z}_1) \\ s, p_s(\boldsymbol{z}_s, \boldsymbol{x}_s \mid \boldsymbol{z}_0, \boldsymbol{z}_1)}} \left[ \sum_{\boldsymbol{x} \neq \boldsymbol{x}_s} u_s^\theta(\boldsymbol{x} \mid \boldsymbol{x}_s) - \sum_{z_s^{(i)} \neq z_1^{(i)}} \frac{\dot{\kappa}_s}{1 - \kappa_s} \log u_s^\theta\big(\boldsymbol{x}(\boldsymbol{z}_s, i, z_1^{(i)}) \mid \boldsymbol{x}_s\big) \right], \tag{5}$$

where $\boldsymbol{x}(\boldsymbol{z}_s, i, z_1^{(i)}) = \text{f}_{\text{rm-blanks}}((z_s^{(1)}, \ldots, z_s^{(i-1)}, z_1^{(i)}, z_s^{(i+1)}, \ldots, z_s^{(n)}))$.

## 3 METHOD

We introduce EDITPP, an Edit Flow process for TPPs that directly learns the joint distribution of event times. Our process leverages the three elementary edit operations *insert*, *substitute*, and *delete* to define a CTMC that continuously interpolates between two event sequences $\boldsymbol{t}_0 \sim p_{\text{noise}}(\boldsymbol{t})$ and $\boldsymbol{t}_1 \sim q_{\text{data}}(\boldsymbol{t})$.

Let $\mathcal{T} = [0, T]$ denote the support of the TPP. We define the state space as $\mathcal{X}_\mathcal{T} = \bigcup_{n=0}^\infty \left\{ (0, t^{(1)}, \ldots, t^{(n)}, T) \in \mathcal{T}^n : 0 < t^{(1)} < \cdots < t^{(n)} < T \right\}$, denoting the set of all possible *padded* TPP sequences with finitely many events. Note that the padding values are introduced for notational simplicity when defining the edit operations on $\mathcal{T}$.

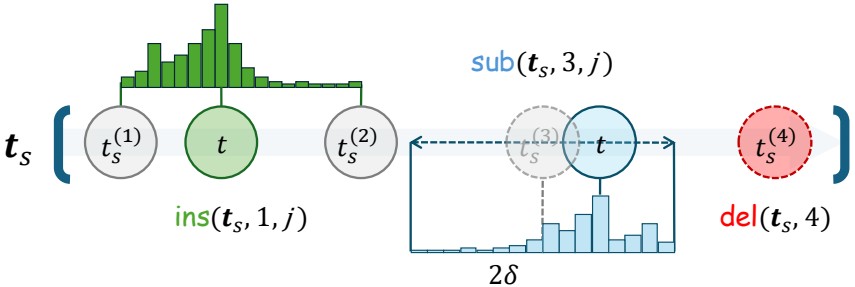

Figure 2: Our discrete edit operations transform continuous event sequences through insertions, substitutions and deletion.

## 3.1 Edit Operations

Our model navigates the state space $\mathcal{X}_\mathcal{T}$ through a set of atomic edit operations. While Edit Flow was originally defined for discrete state spaces, we can generalize the method to continuous state spaces provided that the set of edit operations remains discrete. We achieve this by defining a finite set of edit operations on our continuous state space $\mathcal{X}_\mathcal{T}$ that nonetheless allow us to transition from any sequence $\boldsymbol{t}$ to any other $\boldsymbol{t}'$ through repeated application.

Similar to Havasi et al. (2025), we design our operations to be mutually exclusive: if two sequences differ by exactly one edit, the responsible operation is uniquely determined. This simplifies the parameterization of the model and computation of the Bregman divergence in Eq. (5).

**Insertion**: To discretize the event insertion, we quantize the space between any two adjacent events $t^{(i)}$ and $t^{(i+1)}$ into $b_{\text{ins}}$ evenly-spaced bins. Then, we define the insertion operation relative to the $i$th event as

$$\text{ins}(\boldsymbol{t}, i, j) = \left( t^{(0)}, \ldots, t^{(i)}, t^{(i)} + \frac{j-1+\alpha}{b_{\text{ins}}}(t^{(i+1)} - t^{(i)}), t^{(i+1)}, \ldots, t^{(n+1)} \right) \quad (6)$$

for $i \in \{0, \ldots, n\}$, $j \in [b_{\text{ins}}]$, where $\alpha \sim \mathcal{U}(0,1)$ is a dequantization factor inspired by uniform dequantization in likelihood-based generative models (Theis et al., 2016). The boundary elements $t^{(0)} = 0$ and $t^{(n+1)} = T$ ensure that insertions are possible across the entire support $\mathcal{T}$. Since the bins between different $i$ are non-overlapping, insertions are mutually exclusive.

**Substitution**: We implement event substitutions by discretizing the continuous space around each event into $b_{\text{sub}}$ bins. In this case, the bins are free to overlap, since a substitution is always uniquely determined by the substituted event. We choose a maximum movement distance $\delta$ and define

$$\text{sub}(\boldsymbol{t}, i, j) = \text{sort}\left( \{t^{(0)}, \ldots, t^{(i-1)}, t^{(i+1)}, \ldots, t^{(n+1)}\} \cup \left\{ \tilde{t}^{(i)} \right\} \right) \quad (7)$$

for $i \in \{1, \ldots, n\}$, $j \in [b_{\text{sub}}]$, where $\tilde{t}^{(i)} = \left[ t^{(i)} - \delta + \frac{j-1+\alpha}{b_{\text{sub}}} 2\delta \right]_0^T$ is the updated event restricted to the support $\mathcal{T}$ and, again, $\alpha \sim \mathcal{U}(0,1)$ is a uniform dequantization factor within the $j$-th bin.

**Deletion**: Finally, we define removing event $i \in \{1, \ldots, n\}$ straightforwardly as

$$\text{del}(\boldsymbol{t}, i) = (t^{(0)}, \ldots, t^{(i-1)}, t^{(i+1)}, \ldots, t^{(n+1)}). \quad (8)$$

In combination, these operations facilitate any possible edit of an event sequence through insertions and deletions with substitutions as a shortcut for local delete-insert pairs. Note that we neither allow inserting after the last boundary event nor substituting or deleting the first or last boundary events, thus guaranteeing operations to stay in the state space $\mathcal{X}_\mathcal{T}$. We illustrate the edit operations in Fig. 2.

Our choice of ins, sub and del ensures three key properties: (i) the resulting event sequences remain valid TPPs, (ii) the number of valid operations, e.g. $\text{ins}(\boldsymbol{t}, i, j)$, is independent of the position $i$, which is necessary for efficient parameterization, and (iii) at most one unique operation can transition between any two states, which significantly reduces the complexity of the training loss in Section 3.3. While these properties are comparably simple to achieve for token sequences in language modeling (Havasi et al., 2025), where any token can replace any other, they require special care in the case of

TPPs. del and sub are defined to ensure that the resulting event sequence remains in increasing order and that the padding events $t^{(0)}$ and $t^{(n+1)}$ remain in place. del transitions are unique because the removed event determines exactly which deletion occurred. Similarly, sub transitions are unique because the original position of the substituted event disambiguates the operation, even though two distinct sub operations may yield the same substituted event value. To achieve uniqueness for ins, the insertion bins corresponding to $\mathrm{ins}(t, i, j)$ have to be mutually disjoint for any $i, j$ since insertions lack a removed event to disambiguate them. We achieve this by sizing the bins relative to the distance between $t^{(i)}$ and $t^{(i+1)}$.

**Parameterization**  Generating a new event sequence in the Edit Flow framework then means to emit a continuous stream of edit operations by integrating a rate model $u_s^\theta(\cdot \mid t)$ from $s = 0$ to $s = 1$. The emitted operations transform a noise sequence $t_0$ into a data sample $t_1$ by transitioning through a series of intermediate states $t$. Given a current state $t_s$, we parameterize the transition rates as

$$u_s^\theta(\mathrm{ins}(t_s, i, j) \mid t_s) = \lambda_{s,i}^{\mathrm{ins}}(t_s)\, Q_{s,i}^{\mathrm{ins}}(j \mid t_s), \qquad (9)$$

$$u_s^\theta(\mathrm{sub}(t_s, i, j) \mid t_s) = \lambda_{s,i}^{\mathrm{sub}}(t_s)\, Q_{s,i}^{\mathrm{sub}}(j \mid t_s), \qquad (10)$$

$$u_s^\theta(\mathrm{del}(t_s, i) \mid t_s) = \lambda_{s,i}^{\mathrm{del}}(t_s), \qquad (11)$$

where $\lambda_{s,i}^{\mathrm{del}}, \lambda_{s,i}^{\mathrm{ins}}, \lambda_{s,i}^{\mathrm{sub}}$ denote the total rate of each of the three basic operations at each event $t^{(i)}$. The distributions $Q_{s,i}^{\mathrm{ins}}$ and $Q_{s,i}^{\mathrm{sub}}$ are *categorical* distributions over the discretization bins $j \in [b_{\mathrm{ins}}]$ and $j \in [b_{\mathrm{sub}}]$, respectively. They distribute the total insertion and substitution rates between the specific options.

## 3.2  Auxiliary Alignment Space

Training our rate model $u_s^\theta$ by directly matching a marginalized conditional rate $u_s(t \mid t_1, t_0)$ generating a $p_s(t \mid t_1, t_0)$, as is common in discrete flow matching (Campbell et al., 2024; Gat et al., 2024), is challenging or even intractable for Edit Flows, since it would require accounting for all possible edits that could produce $t$ (Havasi et al., 2025).

To address this, following Havasi et al. (2025), we introduce an auxiliary alignment space for TPPs, where every possible edit operation is uniquely defined in the element wise mixture path $z_s \sim p_s(z_s \mid z_0, z_1)$, making the learning problem tractable.

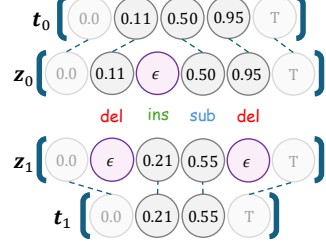

In language modeling, any token can appear in any position, so Havasi et al. (2025) achieve strong results even when training with a simple alignment that juxtaposes two sequences after shifting one of them by a constant number of places. In our case, for the alignments to correspond to possible edit operations, two events can only be matched, i.e., $z_0^{(i)} \neq \epsilon$ and $z_1^{(i)} \neq \epsilon$, if $|z_0^{(i)} - z_1^{(i)}| < \delta$ since otherwise the resulting sub operation would be invalid. Furthermore, $z$s have to correspond to sequences in $\mathcal{X}_\mathcal{T}$, so $f_{\mathrm{rm\text{-}blanks}}(z)$ has to be increasing, and in particular any mixing $z_s$ between $z_0$ and $z_1$ needs to be valid, i.e., $z_s \sim p_s(z_s \mid z_0, z_1) \Rightarrow f_{\mathrm{rm\text{-}blanks}}(z_s) \in \mathcal{X}_\mathcal{T}$.

Figure 3: Illustration of the alignment space for $t_0$ and $t_1$.

We find the minimum-cost alignment between the non-boundary events of $t_0$ and $t_1$ with the Needleman-Wunsch algorithm (Needleman & Wunsch, 1970), i.e.,

$$\mathrm{align}(t_0, t_1) = \mathrm{wrap\text{-}boundaries}\Big(\mathrm{Needleman\text{-}Wunsch}\big(t_0^{(1:n)}, t_1^{(1:m)}, c_{\mathrm{ins}}, c_{\mathrm{sub}}, c_{\mathrm{del}}\big)\Big) \qquad (12)$$

and the cost functions

$$c_{\mathrm{sub}}(i, j) = \begin{cases} |t_0^{(i)} - t_1^{(j)}| & \text{if } |t_0^{(i)} - t_1^{(j)}| < \delta \text{ and } t_0^{(i-1)} < t_1^{(j)} < t_0^{(i+1)} \\ \infty & \text{otherwise} \end{cases}$$

$$c_{\mathrm{ins}}(i, j) = \begin{cases} \frac{\delta}{2} & \text{if } t_0^{(i)} < t_1^{(j)} \\ \infty & \text{otherwise} \end{cases} \qquad c_{\mathrm{del}}(i, j) = \begin{cases} \frac{\delta}{2} & \text{if } t_0^{(i)} > t_1^{(j)} \\ \infty & \text{otherwise} \end{cases} \qquad (13)$$

where wrap-boundaries wraps the sequences with aligned boundary events $0$ and $T$. The algorithm builds up the aligned sequences pair by pair. The operations corresponds to adding different pairs to the end of $(z_0, z_1)$, i.e., insertion $(\epsilon, t_1^{(j)})$, deletion $(t_0^{(i)}, \epsilon)$ and substitution $(t_0^{(i)}, t_1^{(j)})$ (see Fig. 3).

We carefully craft the cost functions in Eq. (13), to guarantee that the minimum-cost alignment corresponds to ins, sub and del operations as we define them in Section 3.1. With the $|t_0^{(i)} - t_1^{(j)}| < \delta$ condition in $c_{\text{sub}}$, we ensure that the aligned sequences will never encode a sub operation for two events that are further than $\delta$ apart. The costs for insertions and deletions and the additional condition on $c_{\text{sub}}$ ensure that the aligned sequences are jointly sorted, i.e., for any $i < j$ we have $\max\left(z_0^{(i)}, z_1^{(i)}\right) < \min\left(z_0^{(j)}, z_1^{(j)}\right)$ where $\min$ and $\max$ ignore $\epsilon$ tokens. This means that any interpolated $\boldsymbol{z}_s$ is sorted by construction. The validity of encoded ins and del operations follows immediately.

## 3.3 TRAINING

We train our model $u_s^{\boldsymbol{\theta}}(\cdot \mid \boldsymbol{t}_s)$ by optimizing the Bregman divergence in Eq. (5). This amounts to sampling from a coupling $\pi(\boldsymbol{z}_0, \boldsymbol{z}_1)$ in the aligned auxiliary space and then matching the ground-truth conditional event rates. Note that the coupling $\pi(\boldsymbol{z}_0, \boldsymbol{z}_1)$ is implicitly defined by its sampling procedure: sample $\boldsymbol{t}_0, \boldsymbol{t}_1 \sim \pi(\boldsymbol{t}_0, \boldsymbol{t}_1)$ from a coupling of the noise and data distribution, e.g., the independent coupling $\pi(\boldsymbol{t}_0, \boldsymbol{t}_1) = p(\boldsymbol{t}_0) q(\boldsymbol{t}_1)$, and then align the sequences $\boldsymbol{z}_0, \boldsymbol{z}_1 = \text{align}(\boldsymbol{t}_0, \boldsymbol{t}_1)$. For our choice of operations, the divergence is

$$\mathcal{L} = \mathop{\mathbb{E}}_{\substack{(\boldsymbol{z}_0, \boldsymbol{z}_1) \sim \pi(\boldsymbol{z}_0, \boldsymbol{z}_1) \\ s, p_s(\boldsymbol{z}_s, \boldsymbol{t}_s | \boldsymbol{z}_0, \boldsymbol{z}_1)}} \left[ \sum_{\omega \in \Omega(\boldsymbol{t}_s)} u_s^{\boldsymbol{\theta}}(\omega \mid \boldsymbol{t}_s) - \sum_{z_s^{(i)} \neq z_1^{(i)}} \frac{\dot{\kappa}_s}{1 - \kappa_s} \log u_s^{\boldsymbol{\theta}}\left(\omega\left(z_s^{(i)}, z_1^{(i)}\right) \mid \boldsymbol{t}_s\right) \right], \quad (14)$$

where $\Omega(\boldsymbol{t}_s)$ is the set of all edit operations applicable to $\boldsymbol{t}_s$ and $\omega\left(z_s^{(i)}, z_1^{(i)}\right)$ is the edit operation encoded in the $i$-th position of the aligned sequences $\boldsymbol{z}_s$ and $\boldsymbol{z}_1$. To make it precise, we have

$$\Omega(\boldsymbol{t}_s) = \bigcup \begin{cases} \{\text{ins}(\boldsymbol{t}_s, i, j) \mid i \in \{0\} \cup [n], j \in [b_{\text{ins}}]\} \\ \{\text{sub}(\boldsymbol{t}_s, i, j) \mid i \in [n], j \in [b_{\text{sub}}]\} \\ \{\text{del}(\boldsymbol{t}_s, i) \mid i \in [n]\} \end{cases} \quad (15)$$

and

$$\omega\left(z_s^{(i)}, z_1^{(i)}\right) = \begin{cases} \text{ins}(\boldsymbol{t}_s, i', j') & \text{if } z_s^{(i)} = \epsilon \text{ and } z_1^{(i)} \neq \epsilon, \\ \text{sub}(\boldsymbol{t}_s, i', j') & \text{if } z_s^{(i)} \neq \epsilon \text{ and } z_1^{(i)} \neq \epsilon, \\ \text{del}(\boldsymbol{t}_s, i') & \text{if } z_s^{(i)} \neq \epsilon \text{ and } z_1^{(i)} = \epsilon. \end{cases} \quad (16)$$

$i'$ is the index such that $\text{f}_{\text{rm-blanks}}(\boldsymbol{z}_s)$ maps $z_s^{(i)}$ to $x_s^{(i')}$ with the convention that $\epsilon$ is mapped to the same $i'$ as the last element of $\boldsymbol{z}_s$ before $i$ that is not $\epsilon$. $j'$ is the index of the insertion or substitution bin relative to $x_s^{(i')}$ that $z_1^{(i)}$ falls into.

## 3.4 SAMPLING

Sampling from our model is done by forward simulation of the CTMC from noise $\boldsymbol{t}_0 \sim p_{\text{noise}}(\boldsymbol{t})$ up to $s = 1$. We follow (Havasi et al., 2025; Gat et al., 2024) and leverage their Euler approximation, since exact simulation is intractable. Even though the rates are parameterized per element, sampling multiple edits within a time horizon can be done in parallel. At each step of length $h$, insertions at position $i$ occur with probability $h \lambda_{s,i}^{\text{ins}}(\boldsymbol{t})$ and deletions or substitutions occur with probability $h(\lambda_{s,i}^{\text{del}}(\boldsymbol{t}) + \lambda_{s,i}^{\text{sub}}(\boldsymbol{t}))$. Since they are mutually exclusive the probability of substitution vs deletion is $\lambda_{s,i}^{\text{sub}}(\boldsymbol{t})/(\lambda_{s,i}^{\text{sub}}(\boldsymbol{t}) + \lambda_{s,i}^{\text{del}}(\boldsymbol{t}))$. Lastly, the inserted or substituted events are drawn from the respective distributions $Q$ to update $\boldsymbol{t}_s$. For a short summary of the unconditional sampling step refer to the Euler update step depicted in algorithm Algorithm 1.

---

**Algorithm 1:** Conditional Sampling

**Input:**
  condition $\boldsymbol{t}_1^c = C(\boldsymbol{t}_1)$, noise $\boldsymbol{t}_0 \sim p_{\text{noise}}, h = 1/n_{\text{steps}}$

$(\boldsymbol{z}_0^c, \boldsymbol{z}_1^c) \leftarrow \text{align}(C(\boldsymbol{t}_0), \boldsymbol{t}_1^c)$

**while** $s < 1$ **do**

    **Euler update**
    Sample edits $\omega_s \sim h\, u_s^\theta(\cdot \mid \boldsymbol{t}_s)$
    $\boldsymbol{t}_{s+h} \leftarrow \text{apply } \omega_s \text{ to } \boldsymbol{t}_s$

    **Recondition**
    $\tilde{\boldsymbol{z}}_{s+h}^c \sim p_{s+h}(\cdot \mid \boldsymbol{z}_0^c, \boldsymbol{z}_1^c)$
    $\boldsymbol{t}_{s+h}^c \leftarrow \text{f}_{\text{rm-blanks}}(\tilde{\boldsymbol{z}}_{s+h}^c)$

    **Merge**
    $\boldsymbol{t}_{s+h} \leftarrow C'(\boldsymbol{t}_{s+h}) \cup \boldsymbol{t}_{s+h}^c$

    $s \leftarrow s + h$

**end**

**Return:** forecast trajectory $C'(\boldsymbol{t}_{s=1})$

---

**Conditional sampling.** We can extend the unconditional model to conditional generation given a binary mask on time $c : \mathcal{T} \rightarrow \{0, 1\}$ (e.g., for forecasting, $c(t) = t \leq t_{\text{history}}$). For a sequence $\boldsymbol{t}$, we define the conditioned part $C(\boldsymbol{t}) = \{t \in \boldsymbol{t} : c(t) = 1\}$ and its complement $C'(\boldsymbol{t})$. Then as depicted in algorithm Algorithm 1, for conditional sampling, we can simply enforce the conditional subsequence to follow a noisy interpolation between $\boldsymbol{t}_0^c = C(\boldsymbol{t}_0)$ and $\boldsymbol{t}_1^c = C(\boldsymbol{t}_1)$, while the complement evolves freely in the sampling process.

## 3.5 Model Architecture

For our rate model $u_s^{\boldsymbol{\theta}}(\cdot \mid \boldsymbol{x}_s)$, we adapt the Llama architecture, a transformer widely applied for variable-length sequences in language modeling (Touvron et al., 2023). We employ `FlexAttention` in the Llama attention blocks, which supports variable-length sequences natively without padding (Dong et al., 2024). As a first step, we convert the scalar event sequence $\boldsymbol{x}_s$ into a sequence of token embeddings by applying $\mathrm{MLP}(\mathrm{SinEmb}(x_s^{(i)}/T))$ to each to each event, where MLP refers to a small multi-layer perceptron (MLP) and SinEmb is a sinusoidal embedding (Vaswani et al., 2017). We convert $s$ and $|\boldsymbol{x}_s|$ into two additional tokens in an equivalent way with separate MLPs and prepend them to the embedding sequence, which we then feed to the Llama. Lastly, we apply one more MLP to map the output embedding $\boldsymbol{h}^{(i)}$ of each event to transition rates. In particular, we parameterize

$$\lambda_{s,i}^{\text{ins}} = \exp(\lambda_{\text{M}} \tanh(\boldsymbol{h}_{\text{ins}}^{(i)})), \quad \lambda_{s,i}^{\text{sub}} = \exp(\lambda_{\text{M}} \tanh(\boldsymbol{h}_{\text{sub}}^{(i)})), \quad \lambda_{s,i}^{\text{del}} = \exp(\lambda_{\text{M}} \tanh(\boldsymbol{h}_{\text{del}}^{(i)})), \quad (17)$$

$$\boldsymbol{Q}_{s,i}^{\text{ins}} = \mathrm{softmax}(\boldsymbol{h}_{\boldsymbol{Q},\text{ins}}^{(i)}), \quad \boldsymbol{Q}_{s,i}^{\text{sub}} = \mathrm{softmax}(\boldsymbol{h}_{\boldsymbol{Q},\text{sub}}^{(i)}). \quad (18)$$

We list the values of all relevant hyperparameters in Appendix A.

## 4 Experiments

We evaluate our model on seven real-world and six synthetic benchmark datasets (Omi et al., 2019; Shchur et al., 2020b; Lüdke et al., 2023; 2025). In our experiments, we compare against IFTPP (Shchur et al., 2020a), an autoregressive baseline which consistently shows state-of-the-art performance (Bosser & Taieb, 2023; Lüdke et al., 2023; Kerrigan et al., 2025). We further compare to PSDIFF (Lüdke et al., 2025) and ADDTHIN (Lüdke et al., 2023), given their strong results in both conditional and unconditional settings and their methodological similarity to our approach. All models are trained with five seeds and we select the best checkpoint based on $W_1$-over-$d_{\text{IET}}$ against a validation set. EDITPP, ADDTHIN, and PSDIFF are trained unconditionally but can be conditioned at inference time.[1] We list the full results in Appendix E.3.

For forecasts, we compare predicted and target sequences by three metrics: $d_{\text{Xiao}}$ introduced by Xiao et al. (2017), the mean relative error (MRE) of the event counts and $d_{\text{IET}}$, which compares inter-event times to quantify the relation between events such as burstiness. In unconditional generation, we compare our generated sequences to the test set in terms of maximum mean discrepancy (MMD) (Shchur et al., 2020b) and their Wasserstein-1 distance with respect to their counts ($d_l$) and inter-event times ($d_{\text{IET}}$). See Appendix C for details.

## 4.1 Unconditional Generation

To evaluate how well samples from each TPP model follow the data distribution, we compute distance metrics between 4000 sampled sequences and a hold-out test set. We report the unconditional sampling results in Table 1. EDITPP achieves the best rank in unconditional sampling by strongly matching the test set distribution across all evaluation metrics, outperforming all baselines. The autoregressive baseline IFTPP shows very strong unconditional sampling capability, closely matching and on some dataset and metric combination outperforming the other non-autoregressive baselines ADDTHIN and PSDIFF.

---

[1] To stay comparable, we employ the conditioning algorithm from Lüdke et al. (2025) for ADDTHIN.

Table 1: Unconditional sampling performance. Bold is best, underlined second best. Ranking follows full results in Appendix E.3 and results are grouped if they fall within the std of the best member.

| | | H1 | H2 | NSP | NSR | SC | SR | PG | R/C | R/P | Tx | Tw | Y/A | Y/M |
|---|---|---|---|---|---|---|---|---|---|---|---|---|---|---|
| MMD | IFTPP | 1.6 | **1.2** | 3.2 | 3.9 | **6.7** | **1.2** | 16.2 | **7.5** | 2.0 | 5.0 | 2.6 | 5.8 | **2.9** |
| | ADDTHIN | 2.4 | 1.8 | 3.5 | 15.7 | 24.6 | 2.5 | 4.6 | **63.0** | 10.2 | 4.1 | 4.4 | 11.8 | 3.7 |
| | PSDIFF | 3.3 | 1.8 | **2.0** | 5.9 | 19.8 | 2.4 | 3.2 | 6.5 | **1.0** | 3.8 | 3.4 | 4.1 | 3.4 |
| | EDITPP | **1.1** | **1.2** | **1.7** | 3.5 | 7.7 | **1.0** | **1.4** | 8.2 | 2.4 | 3.1 | 1.3 | 3.7 | 4.0 |
| | | ×10⁻² | ×10⁻² | ×10⁻² | ×10⁻² | ×10⁻² | ×10⁻² | ×10⁻² | ×10⁻³ | ×10⁻² | ×10⁻² | ×10⁻² | ×10⁻² | ×10⁻² |
| $W_{1,d_t}$ | IFTPP | 20.5 | 13.3 | 11.5 | 14.1 | **1.5** | 23.0 | 294.6 | 3.9 | 3.2 | 2.9 | 6.5 | 3.3 | 2.5 |
| | ADDTHIN | 33.3 | 21.8 | 12.8 | 49.0 | 22.7 | 41.8 | 24.5 | 37.0 | 33.6 | **2.3** | 15.5 | 6.0 | **1.6** |
| | PSDIFF | 26.9 | 29.6 | 5.5 | 13.3 | 10.6 | 30.3 | 16.1 | **1.3** | 2.5 | 2.8 | 6.3 | 1.5 | 1.5 |
| | EDITPP | **7.6** | **7.0** | **3.1** | **1.5** | 1.3 | **6.4** | **6.2** | 1.9 | 5.7 | 2.5 | 3.4 | 1.4 | 1.7 |
| | | ×10⁻³ | ×10⁻³ | ×10⁻³ | ×10⁻³ | ×10⁻³ | ×10⁻³ | ×10⁻³ | ×10⁻² | ×10⁻² | ×10⁻² | ×10⁻³ | ×10⁻² | ×10⁻² |
| $W_{1,d_{IET}}$ | IFTPP | 6.3 | 5.8 | 3.2 | 2.3 | 6.5 | **7.1** | 30.3 | 1.8 | 7.1 | 17.4 | 4.9 | 3.2 | 2.8 |
| | ADDTHIN | 6.6 | 7.0 | 3.2 | 3.9 | 15.1 | 9.4 | 8.0 | 5.3 | 20.0 | **8.8** | 5.5 | 3.2 | 2.4 |
| | PSDIFF | 8.6 | 9.9 | **3.0** | 5.1 | 32.6 | 12.8 | 9.0 | 1.6 | **4.3** | 11.1 | 6.7 | 2.4 | 2.3 |
| | EDITPP | **5.3** | **5.5** | 3.1 | **2.2** | **6.4** | 7.0 | **7.5** | **1.4** | 6.0 | 11.1 | 4.6 | 2.5 | 2.3 |
| | | ×10⁻¹ | ×10⁻¹ | ×10⁻¹ | ×10⁻¹ | ×10⁻² | ×10⁻¹ | ×10⁻² | ×10⁻¹ | ×10⁻³ | ×10⁻² | ×10⁻¹ | ×10⁻¹ | ×10⁻¹ |

## 4.2 CONDITIONAL GENERATION (FORECASTING)

Predicting the future given some history window is a fundamental TPP task. For each test sequence, we uniformly sample 50 forecasting windows $[T_0, T], T_0 \in [\Delta T, T - \Delta T]$, with minimal history and forecast time $\Delta T$. While, this set-up is very similar to the one proposed by Lüdke et al. (2023), there are key differences: we do not fix the forecast window and do not enforce a minimal number of forecast or history events. In fact, even an empty history encodes the information of not having observed an event and a TPP should capture the probability of not observing any event in the future.

We report the forecasting results in Table 2. EDITPP shows very strong forecasting capabilities closely matching or surpassing the baselines across most dataset and metric combinations. Even though IFTPP is explicitly trained to auto-regressively predict the next event given its history, it shows overall worse forecasting capabilities compared to the unconditionally trained EDITPP, ADDTHIN and PSDIFF. This again, underlines previous findings (Lüdke et al., 2023), that autoregressive TPPs can suffer from error accumulation in forecasting. Similar to the unconditional setting, PSDIFF (transformer) outperforms ADDTHIN (convolution with circular padding), which showcases the improved posterior and modeling of long-range interactions.

Table 2: Forecasting accuracy up to $T$. Bold is best, underlined second best. Ranking follows full results in Appendix E.3 and results are grouped if they fall within the std of the best member.

| | | PG | R/C | R/P | Tx | Tw | Y/A | Y/M |
|---|---|---|---|---|---|---|---|---|
| $d_{Xiao}$ | IFTPP | 6.0 | 3.9 | 6.3 | 4.7 | **2.6** | 1.8 | 3.4 |
| | ADDTHIN | 2.5 | 8.8 | 7.3 | **4.0** | 2.8 | 1.5 | **2.9** |
| | PSDIFF | **2.4** | 3.2 | **4.8** | 4.4 | 2.6 | 1.5 | 3.0 |
| | EDITPP | 2.5 | 3.4 | **4.9** | 4.5 | 2.7 | 1.5 | 3.0 |
| | | | | | ×10¹ | ×10¹ | | |
| MRE | IFTPP | 38.9 | 7.5 | 3.5 | 3.2 | **2.1** | 3.7 | 3.9 |
| | ADDTHIN | 3.7 | 14.8 | 4.6 | **3.0** | 3.0 | 3.5 | 3.7 |
| | PSDIFF | **3.4** | **3.3** | 3.0 | 11.4 | 2.4 | 3.5 | 9.2 |
| | EDITPP | 3.5 | 3.6 | **2.8** | 12.3 | 2.3 | 3.5 | 9.0 |
| | | ×10⁻¹ | | | ×10⁻¹ | ×10⁻¹ | | ×10⁻¹ | ×10⁻¹ |
| $d_{IET}$ | IFTPP | 4.7 | 6.8 | 14.7 | 1.4 | 2.2 | 5.9 | 3.9 |
| | ADDTHIN | 4.0 | 6.9 | 10.3 | 1.2 | 1.5 | **4.9** | 2.6 |
| | PSDIFF | 4.1 | 6.2 | **9.5** | 1.1 | 1.5 | **4.9** | 2.6 |
| | EDITPP | 4.0 | 6.8 | 10.1 | 1.1 | **1.4** | 5.0 | 2.7 |
| | | ×10⁻¹ | ×10⁻¹ | ×10⁻³ | ×10⁻¹ | | ×10⁻¹ | ×10⁻¹ |

## 4.3 EDIT EFFICIENCY

The sub operation allows our model to modify sequences in a more targeted way when compared to PSDIFF or ADDTHIN, which have to rely on just inserts and deletes.

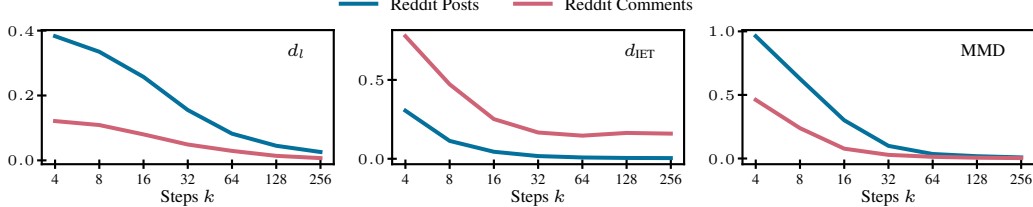

Figure 4: Changing the number of steps $k$ allows trading off compute and sample quality in terms of $d_l$, $d_{\mathrm{IET}}$ and MMD at inference time.

Note that one sub operation can replace an insert-delete pair. Table 3 shows this results in EDITPP using fewer edit operations than PSDIFF on average even if one would count substitutions twice, as an insert and a delete.[2] This is further amplified by the fact, that unlike EDITPP, PSDIFF and ADDTHIN only indirectly parameterizes the transition edit rates by predicting $t_1$ by insertion and deletion at every sampling step.

Table 3: Average number of edit operations in unconditional sampling across datasets. Full result in Table 13.

|         | Ins    | Del   | Sub   | Total  |
|---------|--------|-------|-------|--------|
| PSDIFF  | 173.48 | 61.04 | 0.00  | 234.52 |
| EDITPP  | 137.42 | 33.08 | 29.16 | 199.65 |

In Table 4, we compare their actual sampling run-time for a batch size of 1024 on the two dataset with the longest sequences. Our implementation beats the reference implementations of ADDTHIN and PSDIFF by a large margin. Note, that for a fair comparison, we fixed the number of sampling steps to 100 in all previous evaluations. As a continuous-time model, EDITPP can further trade

Table 4: Sample run-time (ms) on a H100 GPU.

|          | R/P       | R/C       |
|----------|-----------|-----------|
| ADDTHIN  | 18,075.62 | 17,689.36 |
| PSDIFF   | 7,776.35  | 3,913.78  |
| EDITPP   | 4,120.38  | 1,505.68  |

off compute against sample quality at inference time without retraining, in contrast to discrete-time models like ADDTHIN and PSDIFF. Fig. 4 shows that sample quality improves as we increase the number of sampling steps and therefore reduce the discretization step size of the CTMC dynamics. At the same time, the figure also shows rapidly diminishing quality improvements, highlighting potential for substantial speedups with only minor quality loss.

## 5    RELATED WORK

The statistical modeling of TPPs has a long history (Daley & Vere-Jones, 2007; Hawkes, 1971). Classical approaches such as the Hawkes process define parametric conditional intensities, but their limited flexibility has motivated the development of neurally parameterized TPPs:

**Autoregressive Neural TPP**: Most neural TPPs adopt an autoregressive formulation, modeling the distribution of each event conditional on its history. These models consist of two components: a *history encoder* and an *event decoder*. *Encoders* are typically implemented using recurrent neural networks (Du et al., 2016; Shchur et al., 2020a) or attention mechanisms (Zhang et al., 2020a; Zuo et al., 2020; Mei et al., 2022), with attention-based models providing longer-range context at the cost of higher complexity (Shchur et al., 2021). Further, some propose to encode the history of a TPP in a continuous latent stochastic processes (Chen et al., 2020; Enguehard et al., 2020; Jia & Benson, 2019; Hasan et al., 2023). For the *decoder*, a wide variety of parametrizations have been explored. Conditional intensities or related measures (e.g., hazard function or conditional density), can be modeled, parametrically (Mei & Eisner, 2017; Zuo et al., 2020; Zhang et al., 2020a), via neural networks (Omi et al., 2019), mixtures of kernels (Okawa et al., 2019; Soen et al., 2021; Zhang et al., 2020b) and mixture distributions (Shchur et al., 2020a). Generative approaches further enhance flexibility: normalizing flow-based (Shchur et al., 2020b), GAN-based (Xiao et al., 2017), VAE-based (Li et al., 2018), and diffusion-based decoders (Lin et al., 2022; Yuan et al., 2023) have all been

---

[2]Due to its recursive definition, ADDTHIN inserts and subsequently deletes some noise events during sampling, which results in additional edit operations compared to PSDIFF.

proposed. While expressive, autoregressive TPPs are inherently sequential, which makes sampling scale at least linearly with sequence length, can lead to error accumulation in multi-step forecasting and limit conditional generation to forecasting.

**Non-autoregressive Neural TPPs**: Similar to our method, these approaches model event sequences through a latent variable process that refines the entire sequence jointly. Diffusion-inspired (Lüdke et al., 2023; 2025) and flow-based generative models (Kerrigan et al., 2025) have recently emerged as promising alternatives to auto-regressive TPP models by directly modelling the joint distribution over event sequences.

## 6 CONCLUSION

We have presented EDITPP, an Edit Flow for TPPs that generalises diffusion-based set interpolation methods (Lüdke et al., 2023; 2025) with a continuous-time flow model introducing substitution as an additional edit operation. By parameterizing insertions, deletions, and substitutions within a CTMC, our approach enables efficient and flexible sequence modeling for TPPs. Empirical results demonstrate that EDITPP matches state-of-the-art performance in both unconditional and conditional generation tasks across synthetic and real-world datasets, while reducing the number of edit operations.

## ACKNOWLEDGMENTS

We want to thank the Munich Center for Machine Learning for providing compute resources.

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

# A    MODEL PARAMETERS

Table 5: Hyperparameters of our $u_s^{\boldsymbol{\theta}}(\cdot \mid \boldsymbol{x}_s)$ model shared across all datasets.

| Parameter | Value |
|---|---|
| Number of ins bins $b_{\mathrm{ins}}$ | 64 |
| Number of sub bins $b_{\mathrm{sub}}$ | 64 |
| Maximum sub distance $\delta$ | $T/100$ |
| Maximum log-rate $\lambda_{\mathrm{M}}$ | 32 |
| $\kappa(s)$ | $1 - \cos\left(\frac{\pi}{2}s\right)^2$ |
| **Llama architecture:** | |
|   Hidden size $H$ | 64 |
|   Layers | 2 |
|   Attention heads | 4 |
| Optimizer | Adam |
| Sample steps | 100 |

All MLPs have input and output sizes of $H$, except for the final MLP whose output size is determined by the number of $\lambda$ and $Q$ parameters of the rate. The MLPs have a single hidden layer of size $4H$. The sinusoidal embeddings map a scalar $s \in [0, 1]$ to a vector of length $H$. In contrast to Havasi et al. (2025), we choose a cosine $\kappa$ schedule $\kappa(s) = 1 - \cos\left(\frac{\pi}{2}s\right)^2$ as proposed by Nichol & Dhariwal (2021) for diffusion models as it improved results slightly compared $\kappa(s) = s^3$.

For evaluation, we use an exponential moving average (EMA) of the model weights. We also use low-discrepancy sampling of $s$ in Eq. (14) during training to smooth the loss and thus training signal (Kingma et al., 2023; Lienen et al., 2025).

We train all models for 20 000 steps and select the best checkpoint by its $W_1$-over-$d_{\mathrm{IET}}$, which we evaluate on a validation set every 1000 steps.

# B    DATA

## B.1    SYNTHETIC DATASETS

The six synthetic datasets were generated by Shchur et al. (2020b) following the simulation procedures detailed in Section 4.1 of Omi et al. (2019). Each dataset contains 1,000 sequences supported on the interval $T = [0, 100]$. They cover a diverse set of temporal dynamics, defined as follows:

**Hawkes Processes (H1, H2).**    Hawkes processes capture self-exciting features of temporal point processes. The two Hawkes processes are parameterized as follows:

$$\lambda(t \mid \mathcal{H}_t) = \mu + \sum_{t_i < t}\sum_{j=1}^{M} \alpha_j \beta_j \exp\{-\beta_j(t - t_i)\},$$

with **H1** ($M = 1$, $\mu = 0.2$, $\alpha_1 = 0.8$, $\beta_1 = 1.0$) and **H2** ($M = 2$, $\mu = 0.2$, $\alpha_1 = 0.4$, $\beta_1 = 1.0$, $\alpha_2 = 0.4$, $\beta_2 = 20.0$).

**Non-stationary Poisson Process (NSP).**    A periodic time-varying intensity:

$$\lambda(t \mid \mathcal{H}_t) = 0.99 \sin\left(\frac{2\pi t}{20000}\right) + 1.$$

**Stationary Renewal Process (SR).**    Inter-event times $\tau_i = t_{i+1} - t_i$ are i.i.d. from a log-normal distribution (mean 1.0, std. 6.0): this produces bursty patterns with short activity bursts followed by long silent periods.

Table 6: Summary statistics for all synthetic and real-world datasets. $\tau$ is the average inter-event time.

| Full Name | Abbrev. | # Seq. | Mean Len. | Support [0, T] | $\tau$ |
|---|---|---|---|---|---|
| Hawkes 1 | H1 | 1000 | 95.4 | 100 | $1.01 \pm 2.38$ |
| Hawkes 2 | H2 | 1000 | 97.2 | 100 | $0.98 \pm 2.56$ |
| Nonstationary Poisson | NSP | 1000 | 100.3 | 100 | $0.99 \pm 2.22$ |
| Nonstationary Renewal | NSR | 1000 | 98.0 | 100 | $0.98 \pm 1.83$ |
| Self-Correcting | SC | 1000 | 100.2 | 100 | $0.99 \pm 0.71$ |
| Stationary Renewal | SR | 1000 | 109.2 | 100 | $0.83 \pm 2.76$ |
| PUBG | PG | 3001 | 76.5 | 38 minutes | $0.41 \pm 0.56$ |
| Reddit Comments | R/C | 1356 | 295.7 | 24 hours | $0.07 \pm 0.28$ |
| Reddit Submissions | R/P | 1094 | 1129.0 | 24 hours | $0.02 \pm 0.03$ |
| Taxi Pick-ups (Manhattan) | Tx | 182 | 98.4 | 24 hours | $0.24 \pm 0.40$ |
| Twitter Activity | Tw | 2019 | 14.9 | 24 hours | $1.26 \pm 2.80$ |
| Yelp Check-ins (Airport) | Y/A | 319 | 30.5 | 24 hours | $0.77 \pm 1.10$ |
| Yelp Check-ins (Mississauga) | Y/M | 319 | 55.2 | 24 hours | $0.43 \pm 0.96$ |

**Non-stationary Renewal Process (NSR).**    A stationary renewal process is first generated using a gamma distribution (mean 1.0, std. 0.5), then timestamps are time-warped by

$$t'_i = \int_0^{t_i} r(s)\, ds, \qquad r(t) = 0.99 \sin\left(\frac{2\pi t}{20000}\right) + 1.$$

This induces temporally varying expected inter-event intervals while preserving local correlations.

**Self-correcting Process (SC).**    The intensity grows with the time elapsed since the last event:

$$\lambda(t \mid \mathcal{H}_t) = \exp\left(t - \sum_{t_i < t} 1\right).$$

This discourages extended silent periods and promotes regular spacing.

### B.2   REAL-WORLD DATASETS

We use the seven real-world datasets proposed by (Shchur et al., 2020b):

**PG (PUBG)** represents death-event timestamps from matches of PUBG. **R/C (Reddit-Comments)** consists of comment timestamps within the first 24 hours of threads posted on `r/askscience`, covering 01.01.2018–31.12.2019. **R/P (Reddit-Submissions)** captures daily submission timestamps from `r/politics`, covering 01.01.2017–31.12.2019. **Tx (Taxi)** are taxi pick-up events in the southern part of Manhattan, New York. **Tw (Twitter)** covers tweet timestamps of user ID 25073877, collected over multiple years. **Y/A (Yelp-Airport)** consists of check-in events at McCarran International Airport (27 users, year 2018). Lastly, **Y/M (Yelp-Mississauga)** presents check-ins for businesses in the city of Mississauga (27 users, year 2018).

## C   METRICS

A standard way in generative modeling to compare generated and real data is the Wasserstein distance (Heusel et al., 2017). It is the minimum average distance between elements of the two datasets under the optimal (partial) assignment between them,

$$W_p(\mathcal{X}, \mathcal{X}') = \left( \min_{\gamma \in \Gamma(\mathcal{X}, \mathcal{X}')} \mathbb{E}_{(\boldsymbol{x}, \boldsymbol{x}') \sim \gamma} \left[ d(\boldsymbol{x}, \boldsymbol{x}')^p \right] \right)^{1/p} \tag{19}$$

where $d$ is a distance that compares elements from the two sets. In the case of sequences of unequal length, one can choose $d$ itself as a nested Wasserstein distance (Lienen et al., 2024). Xiao et al. (2017) were the first to design such a distance between TPPs. They exploit a special case of $W_1$

for sorted sequences of equal length and assign the remaining events of the longer sequence to pseudo-events at $T$ to define

$$d_{\text{Xiao}}(\boldsymbol{x}, \boldsymbol{x}') = \sum_{i=1}^{|\boldsymbol{x}|} |t^{(i)} - t'^{(i)}| + \sum_{i=|\boldsymbol{x}|+1}^{|\boldsymbol{x}'|} |T - t'^{(i)}| \tag{20}$$

where $\boldsymbol{x}'$ is assumed to be the longer sequence. $d_{\text{Xiao}}$ captures a difference in both location and number of events between two sequences through its two terms.

(Shchur et al., 2020b) propose to compute the MMD between sets based on a Gaussian kernel and $d_{\text{Xiao}}$. In addition, we evaluate the event count distributions via a Wasserstein-1 distance with respect to a difference in event counts $W_{1,d_l}$ where $d_l(\boldsymbol{x}, \boldsymbol{x}') = \big| |\boldsymbol{x}| - |\boldsymbol{x}'| \big|$. Finally, we the distributions of inter-event times between our generated sequences and real sequences in $W_{1,d_{\text{IET}}}$, i.e., a Wasserstein-1 distance of $d_{\text{IET}}$. $d_{\text{IET}}$ is itself the $W_2$ distance between inter-event times of two sequences and quantifies how adjacent events relate to each other to capture more complex patterns.

## D ABLATIONS

We ablate the hyperparameters $\delta$, $b_{\text{ins}}$ and $b_{\text{sub}}$ in Figs. 5 to 8.

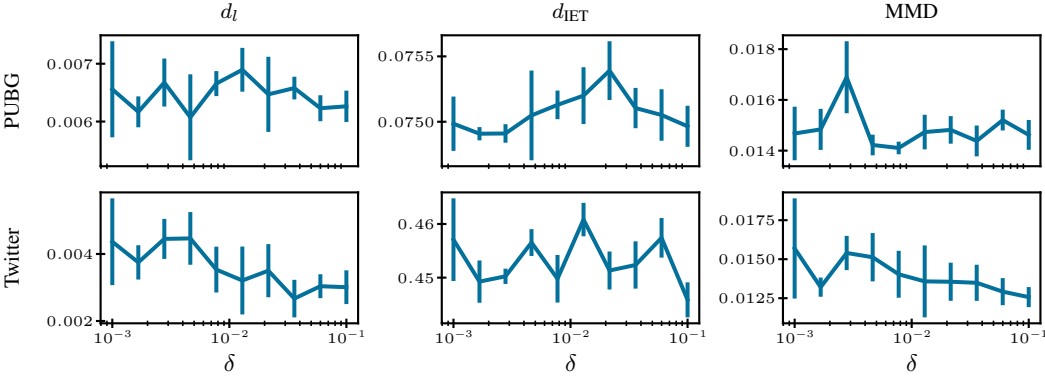

Figure 5: Mean and standard error of $d_l$, $d_{\text{IET}}$ and MMD on two datasets as we vary the $\delta$ parameter for substitutions.

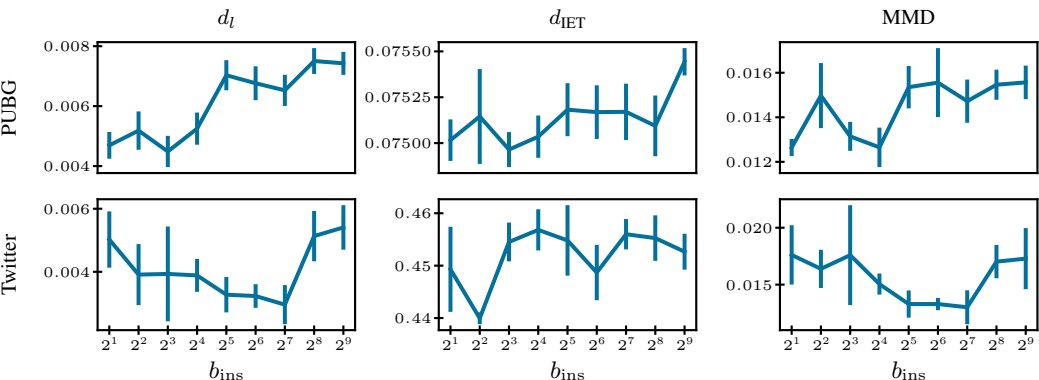

Figure 6: Mean and standard error of $d_l$, $d_{\text{IET}}$ and MMD on two datasets as we vary the number of insertion bins $b_{\text{ins}}$.

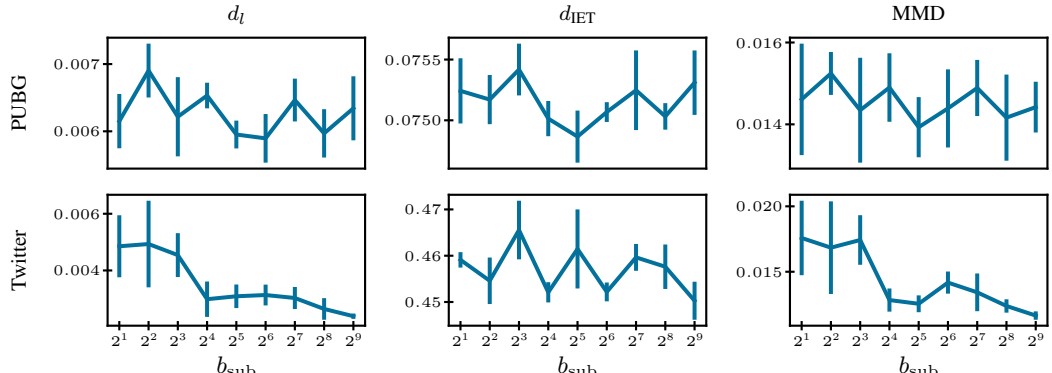

Figure 7: Mean and standard error of $d_l$, $d_{\text{IET}}$ and MMD on two datasets as we vary the number of substitution bins $b_{\text{sub}}$.

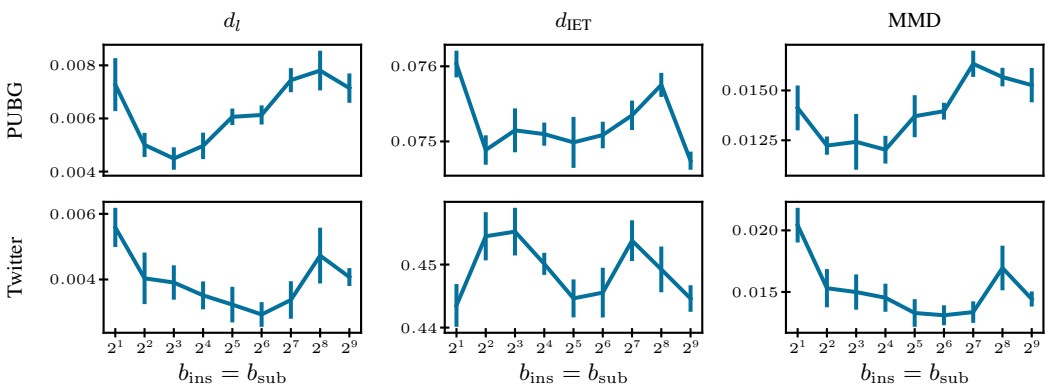

Figure 8: Mean and standard error of $d_l$, $d_{\text{IET}}$ and MMD on two datasets as we vary the number of insertion and substitution bins together.

# E    DETAILED RESULTS

## E.1    INPAINTING VS. FORECASTING

To demonstrate the flexibility of EDITPP for conditional generation, we evaluate its performance when generating events on the interval $[T/3,\ 2T/3]$. In the *forecasting* setting, the model is conditioned only on events occurring before $T/3$, whereas in the *inpainting* setting, it is conditioned on both the past ($t < T/3$) and the future ($t > 2T/3$). As the results show, providing both past and future context substantially improves the quality of the generated middle segment compared to conditioning on the past alone.

|  | PG | R/C | R/P | Tx | Tw | Y/A | Y/M |
|---|---|---|---|---|---|---|---|
| $d_{\text{xiao}}$ |  |  |  |  |  |  |  |
| Inpainting | 2.22 ± 0.04 | 22.66 ± 0.77 | 13.13 ± 0.35 | 3.02 ± 0.20 | 1.58 ± 0.03 | 0.59 ± 0.02 | 1.11 ± 0.04 |
| Forecasting | 2.27 ± 0.05 | 25.53 ± 1.16 | 18.09 ± 0.81 | 3.27 ± 0.33 | 1.65 ± 0.04 | 0.63 ± 0.03 | 1.13 ± 0.05 |
| MRE |  |  |  |  |  |  |  |
| Inpainting | 0.29 ± 0.01 | 5.79 ± 0.96 | 0.20 ± 0.01 | 0.60 ± 0.22 | 1.96 ± 0.09 | 0.48 ± 0.01 | 0.74 ± 0.11 |
| Forecasting | 0.30 ± 0.01 | 5.07 ± 1.13 | 0.33 ± 0.01 | 0.81 ± 0.13 | 2.05 ± 0.11 | 0.52 ± 0.05 | 0.78 ± 0.08 |
| $d_{\text{IET}}$ |  |  |  |  |  |  |  |
| Inpainting | 0.39 ± 0.01 | 0.40 ± 0.01 | 0.01 ± 0.00 | 0.10 ± 0.01 | 1.13 ± 0.02 | 0.80 ± 0.05 | 0.70 ± 0.06 |
| Forecasting | 0.41 ± 0.01 | 0.44 ± 0.03 | 0.02 ± 0.00 | 0.10 ± 0.00 | 1.18 ± 0.01 | 0.78 ± 0.07 | 0.72 ± 0.08 |

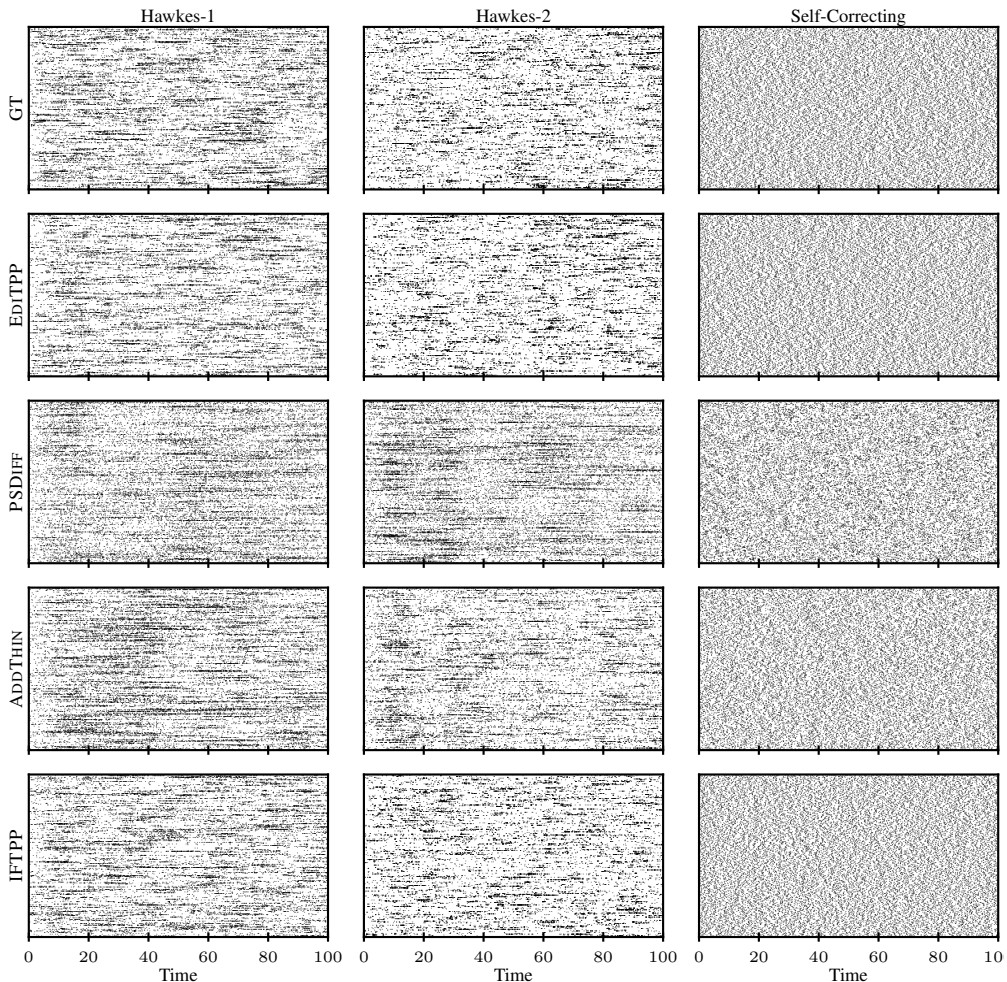

Figure 9: Event times for 200 samples from ground truth data (GT) and each model. Each event sequence is represented as a separate row.

## E.2 PARAMETRIC TPP SAMPLES

To illustrate how well each model captures parametric TPPs, we draw 200 samples for the Hawkes and Self-Correcting processes. In Fig. 10, we plot the cumulative count $N(t)$ for each sample, while Fig. 9 shows each event sequence as a separate row, directly visualizing the events over time. These visualizations further highlight the strong unconditional sampling performance of EDITPP demonstrated in Table 1.

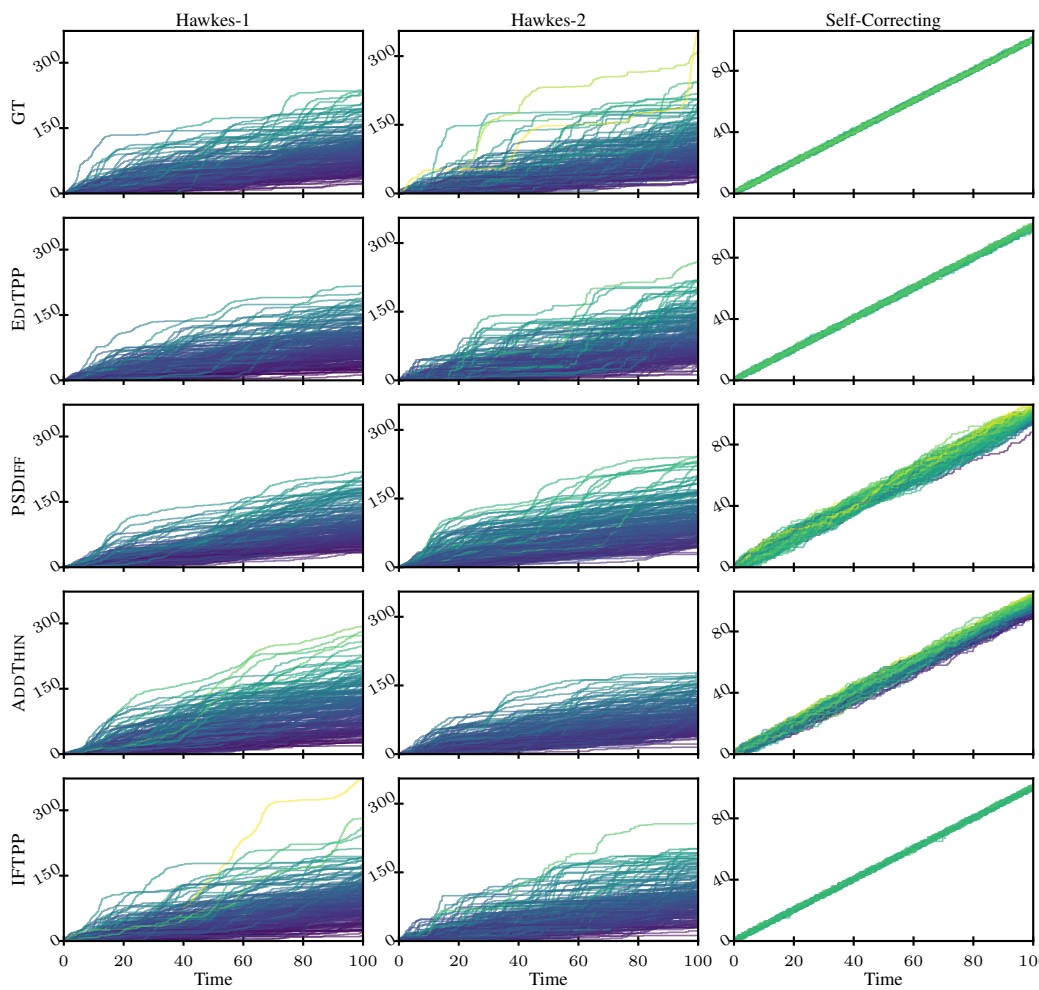

Figure 10: $N(t)$ for 200 samples from ground truth data (GT) and each model.

### E.3 FULL RESULTS

Table 7: Forecasting accuracy up to $T$ measured by $d_{\text{IET}}$.

|  | EDITPP | PSDIFF | ADDTHIN | IFTPP |
|---|---|---|---|---|
| PUBG | **0.400 ± 0.002** | 0.413 ± 0.009 | 0.403 ± 0.010 | 0.473 ± 0.019 |
| Reddit Comments | 0.684 ± 0.005 | **0.625 ± 0.012** | 0.693 ± 0.012 | 0.684 ± 0.012 |
| Reddit Posts | 0.010 ± 0.000 | **0.009 ± 0.000** | 0.010 ± 0.001 | 0.015 ± 0.003 |
| Taxi | **0.113 ± 0.003** | **0.113 ± 0.001** | 0.116 ± 0.001 | 0.145 ± 0.009 |
| Twitter | **1.441 ± 0.020** | 1.487 ± 0.012 | 1.493 ± 0.033 | 2.187 ± 0.029 |
| Yelp Airport | 0.497 ± 0.009 | **0.492 ± 0.005** | 0.493 ± 0.013 | 0.587 ± 0.019 |
| Yelp Mississauga | 0.272 ± 0.003 | 0.262 ± 0.003 | **0.260 ± 0.003** | 0.388 ± 0.024 |

Table 8: Forecasting accuracy up to $T$ measured by mean relative error of event counts.

|  | EDITPP | PSDIFF | ADDTHIN | IFTPP |
|---|---|---|---|---|
| PUBG | 0.349 ± 0.001 | **0.339 ± 0.008** | 0.367 ± 0.005 | 3.892 ± 0.035 |
| Reddit Comments | 3.594 ± 0.118 | **3.260 ± 0.268** | 14.777 ± 3.226 | 7.515 ± 2.112 |
| Reddit Posts | **0.281 ± 0.001** | 0.296 ± 0.006 | 0.457 ± 0.065 | 0.352 ± 0.022 |
| Taxi | 1.234 ± 0.036 | 1.140 ± 0.043 | **0.301 ± 0.014** | 0.321 ± 0.018 |
| Twitter | 2.327 ± 0.042 | 2.435 ± 0.106 | 2.984 ± 0.246 | **2.060 ± 0.027** |
| Yelp Airport | 0.350 ± 0.007 | **0.346 ± 0.004** | 0.347 ± 0.014 | 0.366 ± 0.009 |
| Yelp Mississauga | 0.902 ± 0.027 | 0.920 ± 0.033 | **0.374 ± 0.012** | 0.392 ± 0.012 |

Table 9: Forecasting accuracy up to $T$ measured by $d_{\mathrm{Xiao}}$.

|  | EDITPP | PSDIFF | ADDTHIN | IFTPP |
|---|---|---|---|---|
| PUBG | 2.478 ± 0.007 | **2.400 ± 0.007** | 2.466 ± 0.024 | 5.954 ± 0.195 |
| Reddit Comments | 34.135 ± 0.382 | **32.467 ± 0.534** | 87.666 ± 20.184 | 39.010 ± 7.508 |
| Reddit Posts | **48.776 ± 0.355** | 47.829 ± 1.050 | 72.754 ± 12.134 | 63.256 ± 9.695 |
| Taxi | 4.464 ± 0.088 | 4.444 ± 0.076 | **4.032 ± 0.129** | 4.744 ± 0.125 |
| Twitter | 2.669 ± 0.022 | 2.635 ± 0.078 | 2.802 ± 0.132 | **2.557 ± 0.055** |
| Yelp Airport | **1.524 ± 0.013** | 1.512 ± 0.016 | 1.548 ± 0.026 | 1.795 ± 0.015 |
| Yelp Mississauga | 3.027 ± 0.046 | 3.005 ± 0.046 | **2.895 ± 0.039** | 3.430 ± 0.047 |

Table 10: Sample quality as measured by MMD.

|  | EDITPP | PSDIFF | ADDTHIN | IFTPP |
|---|---|---|---|---|
| Hawkes-1 | **0.011 ± 0.002** | 0.033 ± 0.009 | 0.024 ± 0.009 | 0.016 ± 0.002 |
| Hawkes-2 | **0.012 ± 0.001** | 0.018 ± 0.006 | 0.018 ± 0.006 | **0.012 ± 0.001** |
| Nonstationary Poisson | **0.017 ± 0.003** | 0.020 ± 0.005 | 0.035 ± 0.011 | 0.032 ± 0.008 |
| Nonstationary Renewal | **0.035 ± 0.001** | 0.059 ± 0.006 | 0.157 ± 0.084 | 0.039 ± 0.007 |
| PUBG | **0.014 ± 0.001** | 0.032 ± 0.012 | 0.046 ± 0.025 | 0.162 ± 0.010 |
| Reddit Comments | 0.008 ± 0.001 | **0.006 ± 0.002** | 0.063 ± 0.012 | 0.007 ± 0.003 |
| Reddit Posts | 0.024 ± 0.001 | **0.010 ± 0.002** | 0.102 ± 0.004 | 0.020 ± 0.007 |
| Self-Correcting | **0.077 ± 0.004** | 0.198 ± 0.002 | 0.246 ± 0.018 | **0.067 ± 0.011** |
| Stationary Renewal | **0.010 ± 0.002** | 0.024 ± 0.005 | 0.025 ± 0.013 | 0.012 ± 0.002 |
| Taxi | **0.031 ± 0.002** | 0.038 ± 0.005 | 0.041 ± 0.004 | 0.050 ± 0.003 |
| Twitter | **0.013 ± 0.002** | 0.034 ± 0.007 | 0.044 ± 0.012 | 0.026 ± 0.005 |
| Yelp Airport | **0.037 ± 0.002** | 0.041 ± 0.004 | 0.118 ± 0.036 | 0.058 ± 0.002 |
| Yelp Mississauga | 0.040 ± 0.003 | 0.034 ± 0.007 | 0.037 ± 0.006 | **0.029 ± 0.002** |

Table 11: Sample quality as measured by $W_1$-over-$d_{\mathrm{IET}}$.

|  | EDITPP | PSDIFF | ADDTHIN | IFTPP |
|---|---|---|---|---|
| Hawkes-1 | **0.526 ± 0.020** | 0.865 ± 0.035 | 0.655 ± 0.081 | 0.628 ± 0.030 |
| Hawkes-2 | **0.546 ± 0.005** | 0.991 ± 0.038 | 0.703 ± 0.049 | 0.582 ± 0.009 |
| Nonstationary Poisson | 0.306 ± 0.005 | **0.303 ± 0.007** | 0.318 ± 0.015 | 0.317 ± 0.006 |
| Nonstationary Renewal | 0.224 ± 0.006 | 0.511 ± 0.016 | 0.393 ± 0.064 | **0.229 ± 0.027** |
| PUBG | **0.075 ± 0.000** | 0.090 ± 0.001 | 0.080 ± 0.003 | 0.303 ± 0.039 |
| Reddit Comments | **0.144 ± 0.003** | 0.157 ± 0.006 | 0.532 ± 0.014 | 0.176 ± 0.008 |
| Reddit Posts | 0.006 ± 0.000 | **0.004 ± 0.000** | 0.020 ± 0.001 | 0.007 ± 0.001 |
| Self-Correcting | **0.064 ± 0.000** | 0.326 ± 0.003 | 0.151 ± 0.005 | 0.065 ± 0.001 |
| Stationary Renewal | 0.697 ± 0.018 | 1.281 ± 0.049 | 0.941 ± 0.145 | **0.714 ± 0.028** |
| Taxi | 0.111 ± 0.001 | 0.111 ± 0.001 | **0.088 ± 0.003** | 0.174 ± 0.015 |
| Twitter | **0.460 ± 0.004** | 0.672 ± 0.007 | 0.545 ± 0.024 | 0.492 ± 0.023 |
| Yelp Airport | 0.246 ± 0.002 | **0.244 ± 0.004** | 0.316 ± 0.046 | 0.318 ± 0.017 |
| Yelp Mississauga | 0.226 ± 0.003 | **0.225 ± 0.003** | 0.236 ± 0.004 | 0.276 ± 0.017 |

Table 12: Sample quality as measured by $W_1$-over-$d_l$.

|  | EDITPP | PSDIFF | ADDTHIN | IFTPP |
|---|---|---|---|---|
| Hawkes-1 | **0.008 ± 0.001** | 0.027 ± 0.008 | 0.033 ± 0.015 | 0.020 ± 0.004 |
| Hawkes-2 | **0.007 ± 0.001** | 0.030 ± 0.009 | 0.022 ± 0.014 | 0.013 ± 0.003 |
| Nonstationary Poisson | **0.003 ± 0.001** | 0.006 ± 0.001 | 0.013 ± 0.005 | 0.012 ± 0.003 |
| Nonstationary Renewal | **0.001 ± 0.000** | 0.013 ± 0.001 | 0.049 ± 0.022 | 0.014 ± 0.011 |
| PUBG | **0.006 ± 0.000** | 0.016 ± 0.008 | 0.024 ± 0.014 | 0.295 ± 0.007 |
| Reddit Comments | 0.019 ± 0.002 | **0.013 ± 0.003** | 0.370 ± 0.081 | 0.039 ± 0.023 |
| Reddit Posts | 0.057 ± 0.003 | **0.025 ± 0.003** | 0.336 ± 0.045 | 0.032 ± 0.011 |
| Self-Correcting | **0.001 ± 0.000** | 0.011 ± 0.001 | 0.023 ± 0.002 | **0.001 ± 0.001** |
| Stationary Renewal | **0.006 ± 0.002** | 0.030 ± 0.019 | 0.042 ± 0.022 | 0.023 ± 0.005 |
| Taxi | 0.025 ± 0.002 | **0.028 ± 0.004** | **0.023 ± 0.006** | 0.029 ± 0.003 |
| Twitter | **0.003 ± 0.001** | 0.006 ± 0.003 | 0.015 ± 0.008 | 0.007 ± 0.002 |
| Yelp Airport | **0.014 ± 0.002** | **0.015 ± 0.004** | 0.060 ± 0.021 | 0.033 ± 0.003 |
| Yelp Mississauga | 0.017 ± 0.003 | **0.015 ± 0.002** | **0.016 ± 0.003** | 0.025 ± 0.006 |

Table 13: Average number of edit operations during unconditional sampling.

|  | EDITPP | | | PSDIFF | |
|---|---|---|---|---|---|
|  | Ins | Del | Sub | Ins | Del |
| H1 | 55.05 ± 28.6 | 65.93 ± 10.5 | 37.74 ± 10.6 | 92.58 ± 34.3 | 102.50 ± 10.6 |
| H2 | 63.69 ± 32.4 | 71.72 ± 9.6 | 30.98 ± 8.7 | 97.57 ± 38.9 | 101.30 ± 10.2 |
| NSP | 49.59 ± 7.3 | 49.66 ± 7.1 | 50.56 ± 8.2 | 100.14 ± 9.7 | 100.16 ± 9.8 |
| NSR | 42.39 ± 5.8 | 44.23 ± 7.2 | 55.88 ± 7.8 | 97.58 ± 7.5 | 100.51 ± 10.5 |
| PG | 56.69 ± 7.2 | 19.71 ± 4.4 | 19.96 ± 4.8 | 76.32 ± 8.7 | 40.90 ± 6.4 |
| R/C | 247.83 ± 251.3 | 8.69 ± 6.6 | 15.38 ± 7.4 | 274.49 ± 254.3 | 24.45 ± 5.7 |
| R/P | 972.89 ± 300.2 | 0.11 ± 0.3 | 23.74 ± 5.0 | 1109.78 ± 307.1 | 24.56 ± 7.4 |
| SC | 34.93 ± 5.7 | 34.37 ± 6.9 | 66.18 ± 8.6 | 98.95 ± 7.8 | 99.61 ± 10.2 |
| SR | 70.67 ± 21.6 | 64.09 ± 11.3 | 38.00 ± 11.5 | 108.85 ± 31.4 | 101.92 ± 10.9 |
| Tw | 11.83 ± 9.4 | 22.34 ± 4.6 | 3.04 ± 2.3 | 14.37 ± 10.3 | 24.46 ± 5.2 |
| Tx | 96.62 ± 14.4 | 9.35 ± 3.3 | 17.23 ± 4.8 | 97.69 ± 17.4 | 24.37 ± 5.2 |
| Y/A | 29.43 ± 6.4 | 22.64 ± 5.0 | 9.00 ± 3.4 | 30.42 ± 6.3 | 24.30 ± 4.8 |
| Y/M | 54.81 ± 13.8 | 17.14 ± 4.2 | 11.37 ± 3.8 | 56.54 ± 15.5 | 24.45 ± 4.9 |
| Mean | 137.42 | 33.08 | 29.16 | 173.48 | 61.04 |
| Total | | 199.65 | | | 234.52 |

