# OpenReview forum: "Edit-Based Flow Matching for Temporal Point Processes"
_ICLR.cc/2026/Conference — ICLR 2026 Poster_

### Official Review · Reviewer_YiHr · 2025-10-27

**Soundness:** 3
**Presentation:** 3
**Contribution:** 2
**Rating:** 6
**Confidence:** 4

**Summary:**

This paper introduces EDITPP, an Edit Flow framework tailored to Temporal Point Processes (TPPs). TPPs model sequences of random events in continuous time, where both the timing and number of events are stochastic—making them a natural fit for the Edit Flow paradigm. Although the training and sampling procedures are largely adapted from the original Edit Flow with minimal changes, applying the framework to TPPs requires several non-trivial modifications. In particular, the authors reformulate the three discrete edit operations—insertion, substitution, and deletion—to operate over continuous event times, effectively bridging discrete edit dynamics with continuous temporal domains. A LLaMA-based architecture serves as the backbone, and the model is evaluated on a range of unconditional and conditional event generation tasks.

**Strengths:**

The application of Edit Flow to TPP sounds logical and natural.

The adaptation of the framework from operating in a discrete token space to modeling continuous event arrival times is novel, technically sound, and demonstrates a meaningful extension of the original methodology.

Experimental evaluation is extensive.

**Weaknesses:**

> While natural and flexible, this factorization (referring to autoregression) comes with inherent limitations: sampling scales linearly with sequence length, errors can compound in  multi-step generation, and conditional generation is restricted to forecasting tasks.

The authors claimed the above limitations for autoregressive models but I wonder if the current setup would mitigate these. For example, an autoregression can be understood as (in fact, the Edit Flow authors pointed this out in their paper) all left-to-right insertion operations and in this case, edit based methods don't seem to scale better than AR models wrt sequence length. Clarifications for the other two claims are also needed.

It would be very helpful if the authors could elaborate on why autoregressive models are not favored in this setting since both the data and the task of modeling temporal event sequences are inherently autoregressive to me (if this intuition is wrong, please point out).

**Questions:**

1. The term “element-wise mixture path” used in Section 3.2 is never explicitly defined in the paper. Should we assume it refers to the same element-wise mixture path introduced in the original Edit Flow paper? If so, it would be helpful to clarify this connection explicitly.

2. Table 1 states "bold is best; underlined second best", but several bolded values differ numerically (like in row MMD, col R/C: 7.5, 6.5, 8.2). Could the authors clarify the criterion for deciding which results are marked as “best”?

---

> ### Author Response · Authors · 2025-11-20
>
> We thank the reviewer for the thoughtful feedback and positive remarks regarding the novelty and technical soundness of proposing an Edit Flow process for TPPs. Below we address the specific questions and concerns.
>
> **Limitations of Autoregressive TPPs** (W1):
>
> We agree that it is natural to model TPPs autoregressively. While this is theoretically equivalent to the full joint distribution, practical limitations of autoregressive parameterizations exist:
>
> *Sequential sampling*: Autoregressive models require sampling each event conditioned on all prior events. This sequential process does not exploit parallelism. In contrast, non-autoregressive models (EdiTPP, AddThin, PSDiff) apply the edits in parallel and generate the sequence over a fixed number of diffusion or flow steps. EdiTPP further allows adjusting the number of sampling steps at inference time. To follow your analogy of left-to-right insertion: constraining an Edit Flow to this setting would reproduce autoregressive behaviour, but Edit Flows are more general and allow flexible edits beyond strict left-to-right generation.
>
> *Accumulation of error*: In autoregressive models, each sampled event is fixed and conditions the following events, thus errors can propagate to subsequent steps. Non-autoregressive models can edit each event at every step, potentially correcting initial errors.
>
> *Limited conditional generation*: The sequential nature of autoregressive models enforces unidirectional information flow, which naturally supports forecasting but limits other conditional tasks.
>
> **Element-wise mixture path** (Q1)
>
> Please refer to the description of Edit Flows in Section 2.3 of our background section.
>
>
> **Ranking**
>
> We have revised the ranking and addressed it in the general comment in more detail.
>
>
> Thank you again for your valuable feedback and suggestions. If there are any more changes that you believe would improve the paper, please let us know.

---

> > ### Comment · Reviewer_YiHr · 2025-11-24
> >
> > The authors' rebuttal solved my concern. I will raise my score.

---

### Official Review · Reviewer_eGXi · 2025-10-29

**Soundness:** 3
**Presentation:** 3
**Contribution:** 2
**Rating:** 4
**Confidence:** 2

**Summary:**

This paper introduce EDITPP, a new way for modeling temporal point processes (TPPs) using an edit-baesd discrete flow matching approach. The method  not only inserts and deletes events like other models, but also substitutes them. It learns the rates of these edits with a Llama transformer model , and it uses a CTMC framework with a special alignment space to make training possible.
Because training directly is intractable , the approach uses an auxiliary variable Z, which is an "alignment space". For training, it first samples a noise sequence t_0 and a data sequence t_1 and use the Needleman-Wunsch algorithm to create z_0 and z_1. Then, the model is trained to match the ground-truth edit rates that are defined by this Z alignment path. They test this model on many datasets, both real and synthetic, for tasks like generation and forecasting. It shows that adding the "substitute" operation makes the model more efficient, using fewer edits and running faster than old models like PSDIFF. As a result, EDITPP shows very strong state-of-the-art results on most of the tasks

**Strengths:**

1. The work demonstrates high-Quality presentation and clarity. It is well-written and effectively uses diagrams to explain its complex methodology.
2. The evaluation of the work is comprehensive. It considers both synthetic and real-world datasets, and both unconditional generation and conditional forecasting tasks. The model also shows much better improved computational efficiency compared to existing diffusion-based TPP models.
3. The paper successfully applies edit flows from discrete text to the more complex continuous and ordered TPP space. The solution—using an auxiliary alignment space (Z) and leveraging the Needleman-Wunsch algorithm for alignment to define a tractable training objective —is a necessary adaptation of edit flows for it to be compatible with temporal point process data.

**Weaknesses:**

1. Despite being a successful application of edit flow to TPP data, the approach lacks substantial methodological innovation and novelty. There's no essential difference between the approach and the original edit flow and the introduction of the Needleman-Wunsch algorithm for alignment is an innovation specific to TPP data.
2. The evaluation results of EDITPP are weak. For example, in Table 2 for conditional forecasting, EDITPP underperforms the baseline models in the majority of the cases. The author uses bold font to indicate the best performing model and underscore to indicate the second best performing model based on overall ranking in the tables, resulting in multiple bold fonts and underscores in each column. Such presentation of models' overall ranking is redundant, unnecessary and confusing. I strongly advise the author to improve the presentation of Table 2.
3. The proposed approach cannot handle marked TPP data with event categories.
4. Please refer to the questions section.

**Questions:**

1. Is it possible to extend the edit approach to marked TPP data? For example, can the approach be used to model a TPP for each event category which can be used to compose into one marked TPP? This approach follows existing works like neural Hawkes process [a] that model intensity function for each type of event separately. Alternatively, can we introduce another operation that predict or change event category on top of the insertion, substituting, deleting operations to model marked TPP? Extending the work to marked TPP data could significantly improve the novelty and innovation of the work.
2. Can the approach be adapted to interpolation task in the conditional generation setting? It seems to be a strength of the proposed approach over auto-regressive models like IFTPP.

References:

a. Mei, Hongyuan, and Jason M. Eisner. "The neural hawkes process: A neurally self-modulating multivariate point process." Advances in neural information processing systems 30 (2017).

---

> ### Author Response · Authors · 2025-11-20
>
> Thank you for your detailed and constructive review, and for recognizing the quality and clarity of our presentation as well as the comprehensive evaluation showing strong state-of-the-art results on most of the tasks. We address your concerns below.
>
> **Methodological Novelty** (W1)
>
> We want to respectfully clarify both the novelty and intent of the paper. Edit Flows introduced a framework for generative modeling of variable-length discrete sequences (language) by defining discrete edit operations and making training tractable through a (random-overlap) alignment space. As such Edit Flows can not trivially be applied to TPPs. With EdiTPP we show how to attain a well-defined and tractable Edit Flow for continuous valued sequences and make the following key contributions:
>
> 1. **Edit operations for continuous valued TPPs (Section 3.1):**
>     In discrete edit flows, insertions or substitutions can introduce any token anywhere. Besides consisting of continuous sequence element, for TPPs, insertion and substitution must respect local temporal ordering, leading us to define (a) a relative insertion mechanism tied to neighboring events and (b) a continuous substitute kernel that preserves monotonicity.
> 2. **Alignment space for ordered continuous sequences (Section 3.2):**
>     Edit Flows are based on alignment spaces and specifically explored a worse-case and random overlap alignment. One can consider AddThin and PSDiff as a discrete-time Markov Chain edit model with worst-case alignment, since they delete and then insert everything. In turn, EdiTPP is a more flexible CTMC and introduces a locally optimal alignment space using Needleman–Wunsch, which strongly reduces the number of required edits.
> 3. **Parametrization of CTMC edit rates (Section 3.5):**
>     We propose new parameterizations for edit intensities over continuous event sequences. In contrast to AddThin and PSDiff, we directly parameterize the instantaneuous edit rates and do not have to predict every missing event at every diffusion step.
> 4. **Conditional Generation (Section 3.4):**
>     We show how flexible conditioning for our edit flow process can be performed at inference time via a masking mechanism on the event domain and enabling forecasting and other partial-sequence completion tasks.
>
> Thus, while EdiTPP builds on Edit Flows for discrete sequences, the resulting formulation constitutes a new category of Edit Flows operating on continuous sequence elements. In doing so, EdiTPP unifies and generalizes earlier TPP approaches into a more flexible and efficient CTMC framework that incorporates an additional substitute operation and extends edit operations to continuous event-time sequences.
>
>
> **State-of-the-art results** (W2)
>
> EdiTPP performs on par with the state-of-the-art, while being significanly more efficient (about 50% less sample time) and reducing the number of edit operations (c.f. Section 4.3). Furthemore, in contrast to previous method our CTMC-based method allows to explicitly trade-off modelling capacity and compute at inference time (c.f. Figure 4). Additionally, our method visually shows superior performance for parametric TPPs (Appendix E. 2), especially when compared to the other non-autoregressive TPP models (PSDiff and AddThin).
>
>
> **Marks** (W3, Q1)
>
> We agree that including marks could be an interesting extension of our model. In this paper, we focused on proposing an Edit Flow process for continuous valued TPPs subsequently generalizing previous state-of-the-art non-autoregressive TPP models and leave the extension to marks for future work. However, EdiTPP can be extended to marked TPPs and we want to highlight three straighforward extensions:
>
> 1. **Type-specific edit rates (multivariate TPP):**
> As in Neural Hawkes, in EdiTPP one could treat each event type as an independent dimension with its own insert/delete/substitute rates, while sharing the Llama backbone.
> 2. **Introducing a type rate for insertion and substitution:**
> One can model events as a joint variable (t,m) and treat a mark m as a discrete token value and applying the edit operations for tokens from Edit Flows.
> 3. **Modeling p(m|t):**
>     As with other (neural) TPPs, the mark can be modelled conditional on the event time.
>
> **Interpolation and other conditional task** (Q2)
>
> Yes any masking function on the interval (as defined in Section 3.4) can be leverage for conditional generation. For comparability we presented forecasting in our main results, but have added an "inpainting"-like conditional task to the Appendix E. 1. The additional experiment shows that conditioning on both past and future can further improve the conditional generation.
>
> **Ranking**
>
> We have revised the ranking and addressed it in the general comment in more detail.
>
> Thank you again for your valuable feedback and suggestions. If there are any more changes that you believe would improve the paper, please let us know.

---

> ### Comment · Reviewer_eGXi · 2025-11-28
> **Post-rebuttal Acknowledgement**
>
> I would like to thank the reviewer for the detailed rebuttal. I'm still not fully convinced that this work should be accepted.
>
> The author reiterated the **novelties** of their work. They are well motivated and I acknowledge that. I respectfully disagree with the author that the EdiTPP performs on-par with baseline models (e.g. by counting the number of times EdiTPP is the top-performing model or looking at specific metrics like MRE) and the performance gap could be justified with efficiency.
>
> I'm also disappointed that the work doesn't include Marked TPP. Based on the discussions, I believe including Marked TPP experiments could significantly improve the work's completeness.

---

> > ### Author Response · Authors · 2025-12-03
> >
> > We want to thank the reviewer for the ongoing engagement in the rebuttal. We are glad that we were able to address your novelty concern.
> >
> > We would like to emphasize that EdiTPP achieves new state-of-the-art results for unconditional TPP modeling (Table 1), and especially qualitatively outperforms other non-autoregressive baselines (Appendix E.2, Fig. 9). In the conditional setting, EdiTPP outperforms IFTPP on 17/21 metric–dataset combinations, and is either better or within 0.1 of PsDiff and AddThin in 16/21 and 18/21 cases, respectively. Thus, our results show that EdiTPP offers competitive or superior performance while retaining significant efficiency and flexibility advantages.
> >
> > We respectfully disagree on the relevance of discrete marks in the context of TPP modelling. The core difficulty in modeling Temporal Point Processes lies in capturing complex temporal interactions at arbitrary and irregular time points and accurately modeling the underlying counting process. Extending a TPP to include discrete marks is, by contrast, largely an additive layer on top of the temporal backbone: marks are typically modeled via a conditional categorical distribution given the event time, which does not alter the underlying temporal dynamics or continuous-time structure that make TPPs challenging. Hence, none of the established non-autoregressive baselines [1,2] model marks. Adding a marked experiment would therefore require a new evaluation pipeline, including new datasets, new baselines, and full hyperparameter tuning for this setting, while offering very limited additional insight into the temporal modeling contributions of EdiTPP.
> >
> > For these reasons, and to remain consistent with the established benchmark setting, both our baselines and EdiTPP focus solely on the temporal TPP setting.
> >
> > [1] Add and thin: Diffusion for temporal point processes. In Thirty-seventh Conference on Neural Information Processing Systems, 2023.
> >
> > [2] Unlocking point processes through point set diffusion. In The Thirteenth International Conference on Learning Representations, 2025.

---

### Official Review · Reviewer_BAAf · 2025-10-31

**Soundness:** 3
**Presentation:** 3
**Contribution:** 3
**Rating:** 6
**Confidence:** 3

**Summary:**

The paper introduces a novel framework for modeling Temporal Point Processes (TPPs) through continuous-time editing operations.

The paper presents the EDITPP framework, which integrates random set interpolation methods for TPPs with discrete sequence editing flow methods. This combination allows for a more effective modeling of event sequences.
The framework parameterizes instantaneous rates of insertion, deletion, and substitution operations within a Continuous-Time Markov Chain (CTMC) framework. This approach significantly reduces the number of editing operations required during the generation process.
The experimental results demonstrate that EDITPP achieves state-of-the-art performance on various real-world and synthetic datasets for both unconditional and conditional tasks.
By introducing an auxiliary alignment space, the framework can be flexibly applied to tasks of varying complexity, enhancing its versatility in different scenarios.
Overall, the paper showcases the effectiveness of EDITPP in improving the modeling and generation of event sequences in TPPs.

**Strengths:**

1. EDITPP effectively combines random set interpolation methods for Temporal Point Processes (TPPs) with discrete sequence editing flow methods, allowing for a more comprehensive modeling approach.
2. By parameterizing the instantaneous rates of insertion, deletion, and substitution within a Continuous-Time Markov Chain (CTMC) framework, EDITPP significantly reduces the number of editing operations required during the generation process, enhancing efficiency.
3. The framework demonstrates state-of-the-art performance on various real-world and synthetic datasets for both unconditional and conditional tasks, showcasing its effectiveness and reliability in practical applications.
4. The introduction of an auxiliary alignment space allows EDITPP to be flexibly applied to tasks of varying complexity, making it versatile for different modeling scenarios and enhancing its applicability across diverse domains.

**Weaknesses:**

1. While the framework shows strong performance on specific datasets, its generalization capabilities to unseen data or different domains may be limited, necessitating further validation across a broader range of applications.
2. Although EDITPP reduces the number of editing operations, the overall computational efficiency during training and inference may still be a concern, especially when dealing with large-scale datasets or real-time applications.

**Questions:**

1. Could you provide more detailed guidelines or best practices for hyperparameter tuning? Given that the performance of EDITPP may heavily rely on this aspect, clearer instructions could help practitioners achieve optimal results more efficiently.

2. How does EDITPP perform when applied to datasets that differ significantly from those used in training? Are there any specific limitations observed in terms of generalization?

3. What measures have been taken to ensure the computational efficiency of EDITPP during both training and inference? Are there specific scenarios where the model may struggle with efficiency?

---

> ### Author Response · Authors · 2025-11-20
>
> We thank the reviewer for the thoughtful feedback and appreciation of our method. We address the concerns and questions below.
>
> **Generalization** (W1, Q2)
>
> We benchmarked our model on **13 synthetic and real-world datasets** spanning diverse TPP patterns and observed consistently strong performance, demonstrating robustness even without dataset-specific hyperparameter tuning.
>
> **Computational efficiency** (W2, Q3)
>
> Our method is built upon a Llama backbone, commonly used for (large) language modeling. This allows our approach to scale to very large datasets and long context windows while leveraging state-of-the-art attention mechanisms and efficient GPU implementations. To the best of our knowledge, we are the first to use an attention mechanism that does not require padding TPP sequences. Additionally, we show that our model is more efficient than existing baselines in both sampling time and the number of edit operations. Unlike previous models, our approach further allows an explicit trade-off between computation and generative capacity at inference time by adjusting the number of sampling steps, giving practitioners flexibility to balance speed and quality as needed.
>
> **Hyperparameters** (Q1)
>
> Notably, we used the same hyperparameters across all 13 datasets, demonstrating the stability and robustness of our method. We consider the number of sampling steps (Figure 4), the number of bins, and the substitution delta as the main hyperparameters of our model, and provide a detailed hyperparameter study in Appendix D. Thus, while dataset-specific tuning can yield additional improvements, the default settings already perform consistently well.
>
> Thank you again for your valuable feedback and suggestions. If there are any more changes that you believe would improve the paper, please let us know.

---

### Official Review · Reviewer_6JS9 · 2025-10-31

**Soundness:** 2
**Presentation:** 3
**Contribution:** 3
**Rating:** 4
**Confidence:** 4

**Summary:**

This paper proposes a non-autoregressive approach to model temporal point processes (TPP) based on Edit Flows (Havasi, 2025). The proposed method models the rates of edits (insertion, deletion, substitution) via a CTMC. Additionally, it supports modeling sequences of variable lengths by defining an auxiliary alignment space that avoids a separate classifier to determine the number of events during an observation window $[0,T]$.

**Strengths:**

* The TPP notation is clear and consistent throughout the paper, making the math easier to follow.
* The idea is interesting and novel, especially the design of auxiliary alignment space to support varying lengths of TPPs, which has been one of the key challenges for non-autoregressive approaches.

**Weaknesses:**

* The idea closely follows the Edit Flows paper for language modeling, from parameterization to writing, e.g., Eq. 9-11 in this paper and Eq. 13-15 in the Edit Flows paper; Eq. 14 in this paper and Eq. 23 in the Edit Flows paper. The main difference between language modeling and TPP is that language modeling uses _discrete_ tokens, while TPPs have _continuous_ timestamps. It is unclear to me how the proposed method handles this. Note that I do not take this as a concern of novelty, but the gap could be better addressed, e.g., there is undefined notation in this paper (borrowed from Edit Flows). Please see more detailed questions below.
* The experimental section is relatively weak. The dataset details are missing, and the paper only uses abbreviations in the main text. The results are also hard to interpret (i.e., a lot of numbers are in bold/underlined). The main text mentions that the ranking is based on the full results in the appendix, but the full results seem to be mixed.
* The relationship between this paper and other non-autoregressive neural TPP papers is less clear, and the paper lacks visualizations on parametric TPP to verify the methodology, e.g., Hawkes processes.

**Questions:**

I do like the idea, so if the authors could help address the following key aspects in addition to clearer experimental results, I would be happy to support the paper:
1. How did the proposed method adapt the discrete flow in the Edit Flows paper to the continuous variable $t$? More specifically, in line 153, if $n$ goes from 0 to infinity, $t^{(n)}$ could have index 0 and infinity, then how can we have a finite number of events? Could the authors help better explain how the state space is defined, and why it is possible to define the set of _all possible_ padded TPP sequences? What does "padded" refer to?
2. For insertion and substitution, if we use a fixed number of $b_\text{ins}$ and $b_\text{sub}$ per dataset, can the method handle event sequences where the inter-arrival time distribution has a large variance? I do not expect the method to work for all scenarios, but it would be great to better understand the strengths and limitations of the method.
3. For the rate model $u_s^{\theta}$ the paper says they used a Llama backbone (line 309). Is there any specific reason for this modeling choice in the context of TPP?
4. If the interpolated $z_s$ needs to be sorted (line 252-253), does the training take longer?
5. Could the authors elaborate on the "seven real-world and six synthetic benchmark datasets", e.g., which version of the datasets did the paper use, what are the parameters for parametric TPPs, and what are the summary statistics for real-world datasets? It would be better to have the details in a self-contained paper.
6. Would it also be possible to visualize the ground truth and sampled events for some parametric TPPs and compare different methods, e.g., Hawkes processes and self-correcting processes (if I correctly understand the results in Table 1)?

Other comments:
* $\epsilon$ and $\text{f}_{\text{rm-blanks}}$ are notations in the Edit Flows paper, but not defined in this paper, e.g., it is used in line 231.
* Figure 4, the metric $d_{W_2}$ is missing.
* It would be great to clarify the ranking in results, e.g., taking the average of five random seeds, and also add error bounds to Table 3 for the number of edit operations.

Typo:
* Eq.2: $F(T|\mathcal{H_t})$ -> $F(T|\mathcal{H_{t^{(n)}}})$.

---

> ### Author Response · Authors · 2025-11-20
>
> Thank you for the thoughtful review, we appreciate the positive feedback for our novel approach and the constructive suggestions and have addressed each point below:
>
> **State space definition** (W1, Q1)
>
> TPPs on a bounded domain are almost surely finite, with the exception of certain cases such as supercritical Hawkes processes. Our definition of the state space denotes a union over all natural numbers $n$, so $\mathcal{X}_{\mathcal{T}}$ is the union of all possible finite event sequences. The use of left and right padding for event sequences is introduced for notational convenience, allowing us to define insertions on the whole domain, i.e., before the first or after the last event.
>
> We have revised the manuscript to better motivate this definition.
>
> **Fixed number of bins per dataset** (Q2)
>
> We used the same hyperparameters across all datasets and provide a detailed hyperparameter study in Appendix D. Importantly, we did not observe any limitations associated with the fixed bin hyperparameter across the 13 benchmark datasets. Since insertions are made relative to neighboring events, the effective temporal precision naturally adapts to local inter-arrival times. Moreover, the uniform noise ensures local smoothness in the generated sequences. The substitution range, controlled by the delta parameter, allows further fine-grained local adjustments and, within reasonable values, does not strongly impact generation quality. In contrast, the baseline methods rely on mixtures of 8 or 16 Gaussians to model event insertions over larger intervals (e.g., [t, T] or [0, T]).
>
> **Llama backbone for TPPs** (Q3)
>
> Transformer-based encoders are standard in TPP modeling, and the Llama backbone provides a well-established and widely adopted transformer architecture, with highly optimized implementations readily available. Furthermore, we can leverage FlexAttention to handle event sequences without padding, thereby reducing both memory and computation costs.
>
> **Sorting complexity** (Q4)
>
> Our choice of cost functions for Needleman-Wunsch constructs the alignment space in such a way that any sampled $\mathbf{z}_s$ will automatically be sorted in ascending order. So no explicit sorting is required in training and thus there is no impact on training times.
>
> We have updated the phrasing slightly.
>
> **Benchmark datasets** (W2, Q5)
>
> For comparability we have used the benchmark datasets from AddThin and PSDiff. To make the paper more self-contained, we added the requested information to the dataset section in the Appendix B.
>
> **Parametric TPP visualization** (W3, Q6)
>
> We have added examples for the requested Hawkes and self-correcting processes to Appendix E.2. These visualizations highlight that our model can capture the self-exciting property of the Hawkes process and the self-inhibiting property of the self-correcting process, thus further underlining the strong unconditional sampling performance of EdiTPP demonstrated in Table 1. In particular, when compared to the other non-autoregressive TPP models (PSDiff and AddThin), our method shows superior performance.
>
> **Minor points**:
>
> - Epsilon and remove blanks not defined in this paper: We have added the definitions to the background.
> - Updated Figure 4
> - Fixed Typo
> - Added standard deviation for Edit Operations
> - We have clarified the ranking in the global comment.
>
> Thank you again for your valuable feedback and suggestions. If there are any more changes that you believe would improve the paper, please let us know.

---

> ### Comment · Reviewer_6JS9 · 2025-11-26
>
> Thanks the authors for their responses. Regarding the state space definition, my understanding of "finite" refers to the number of events in one sequence defined on $[0,T]$. However, it is still unclear to me how to define the **uncountable set** of event sequences because there are infinite possibilities of event time with continuous timestamps. The "padding" definition is also confusing; does it refer to the start and the end of the event sequences?
>
> The current version, without better addressing the continuous nature of TPPs, does not clarify the novelty of adapting the "Edit Flows" paper other than straightforward discretization. The CTMC, edit operations, and the auxiliary alignment process are proposed and addressed in the "Edit Flows" paper, and the current manuscript lacks sufficient details to reproduce the results. It would greatly help the readers if a revised version could elaborate on the key challenges of adapting Edit Flows and how the proposed approach addresses them.

---

> > ### Author Response · Authors · 2025-11-26
> >
> > Thank you for your continued feedback.
> >
> > To further clarify the definition of the space of all possible TPP sequences: An event sequence $(t^{(0)}, t^{(1)}, \ldots, t^{(n)}, t^{(n + 1)})$ is part of $\mathcal{X}\_{\mathcal{T}}$ if and only if $t^{(0)} < t^{(1)} < \ldots < t^{(n)} < t^{(n + 1)}$ and $t^{(0)} = 0$ and $t^{(n + 1)} = T$ for any $n \in \mathbb{N}\_0$. We refer to $t^{(0)}$ and $t^{(n+1)}$ as padding, because they are constants that we prepend and append to every event sequence. Although the space $\mathcal{X}\_{\mathcal{T}}$ is uncountable, this does not preclude the definition of a generative model as per the existence of generative models on $\mathbb{R}^n$. Moreover, in our model we never need to actually consider all sequences in $\mathcal{X}\_{\mathcal{T}}$. The transition rates are defined only between sequences that differ by at most one event, which keeps all transition rates local in $\mathcal{X}\_{\mathcal{T}}$.
> >
> > We want to emphasize that EdiTPP is not a discretization of Edit Flows, quite the opposite in fact. Edit Flows have been developed for a discrete state space and with EdiTPP we propose a "continuization" of Edit Flows. Of course, the general structure of a CTMC, edit operations and an auxiliary alignment process have been proposed in the Edit Flows paper and we hope to have to made it abundantly clear in our paper that we are building on top of the excellent work by Havasi et al. However, applying these concepts to TPPs is far from trivial. As we show in Section 3.1, designing the insertion and substitution operations to allow the model to navigate the complete state space while simultaneously guaranteeing that any two event sequences can be transformed into each other by at most one unique operation requires great care. Without diminishing Havasi et al. in the slightest, they achieved their inspiring results with a comparably simple construction for their auxiliary alignment process: a constant shift between the token sequences by 50 places. This is not viable for TPPs, because any interpolated sequence $\mathbf{z}_s$ has to both remain sorted in increasing order for any $s$ and any possible mixing of the sequences $z_0$ and $z_1$ and conform to our more complex edit operations. Constructing the cost functions for Needleman-Wunsch in Eq. (13) in Section 3.2, so that the optimal alignment upholds these properties is itself a significant contribution. As you suggested, we have added a paragraph in Sections 3.1 and a clarification in Section 3.2 to help readers understand the key challenges in adapting Edit Flows for TPPs and how we address them.
> >
> > We hope to have addressed your remaining concerns and are happy to discuss any further questions.

---

### Author Response · Authors · 2025-11-20

We thank the reviewers for their constructive and valuable feedback. While we have addressed each point individually, we would like to highlight some additional results and an updated ranking of the results tables.

**Additional results**

We have added a hyperparameter study to Appendix D. Additionally, we have ablated our method to compare forecasting (conditioned only on the past) versus inpainting (conditioned on both past and future) in Appendix E. 1. Finally, we have included sample results for requested parametric TPP datasets in Appendix E. 2.

**Clarified ranking of results**

Previously, we bolded each value whose standard deviation overlaps with the best value, or for which the standard deviation of the best value overlaps with it. Since this scheme was unintuitive for several reviewers, we have updated it: values are now bolded only if they fall within the standard deviation of the best model. Note that the ranking is conducted on the full results in Appendix E.

---

### Author Response · Authors · 2025-12-03

We thank the reviewers for their active engagement throughout the rebuttal process, which helped us further refine and improve the paper. Below, we provide a brief overview of the main points addressed in the discussions:

Reviewer 6JS9:
We clarified the continuous-state edit operations, alignment space, and padding. We added detailed dataset descriptions and visualizations of parametric TPPs and revised the ranking scheme. All follow-up questions were addressed with additional technical explanations.

Reviewer BAAf:
The reviewer raised questions with regards to generalization, efficiency, and hyperparameters. We added a complete hyperparameter study and explained why the method generalizes well and is more efficient than the baselines. No concerns remain.

Reviewer eGXi:
We addressed the questions on novelty and table formatting, clarified all methodological extensions beyond discrete Edit Flows, and updated the ranking. The remaining hesitation concerns marked TPPs, which are not part of the benchmark and are also not modeled by the non-autoregressive baselines.

Reviewer YiHr:
All concerns (autoregressive limitation, definition of the mixture path, and ranking) were fully resolved, and the reviewer explicitly raised their score.

Overall, we addressed all reviewer questions through clarifications, added experiments, and revisions. The paper now reflects a complete and refined version incorporating the full reviewer feedback.

---

### Meta-Review · Area_Chair_hi8L · 2026-01-06

**Summary:**

This paper generalized Edit-Based Flow Matching to TPPs, which necessitate handling continuous time indices.

**Reviewer Concerns:**

While reviewers mostly agreed that the approach was sound and novel (especially after score adjustment), it seemed that the biggest questions were over whether:
1. the exposition was sufficiently clear as to communicate the novelty (Reviewer 6JS9)
2. Whether Edit-TPP outperforms baselines, and the extent to which those baselines are thoroughly explained  (Reviewer 6JS9, eGXi)
I believe that the responses regarding mathematical formulation could have been made more precise, but I suspect that this would have been achieved had the logistics of the review process not been interrupted.

**Reviewer Scores:**

I believe that reveiwer 6JS9 would have raised their score to a 6, given their initial positive inclination towards the paper and the authors responses, possibly to an 8. Reviewer eGXi acknowledged the novelty of the paper, but still had issues about experiments (and wanted study of Marked TPP). It is still possible that they may have raised their score. Similarly it is not clear if BAAf would have raised their score. Hence, it seems a likely outcome would be 6664 or 6666. This puts it at a borderline accept.

---

### Decision · Program_Chairs · 2026-01-26

Accept (Poster)